# LazyDrag: Enabling Stable Drag-Based Editing on Multi-Modal Diffusion Transformers via Explicit Correspondence

**Zixin Yin**[1,2]    **Xili Dai**[3]    **Duomin Wang**[2]
**Xianfang Zeng**[2]    **Lionel M. Ni**[1,3]    **Gang Yu**[2]    **Heung-Yeung Shum**[1]

[1] The Hong Kong University of Science and Technology    [2] StepFun
[3] The Hong Kong University of Science and Technology (Guangzhou)

## Abstract

The reliance on implicit point matching via attention has become a core bottleneck in drag-based editing, resulting in a fundamental compromise on weakened inversion strength and costly test-time optimization (TTO). This compromise severely limits the generative capabilities, suppressing high-fidelity inpainting and text-guided creation. In this paper, we introduce LazyDrag, the first drag-based image editing method for Multi-Modal Diffusion Transformers, which directly eliminates the reliance on implicit point matching. In concrete terms, our method generates an explicit correspondence map from user drag inputs as a reliable reference to boost the attention control. This reliable reference opens the potential for a stable full-strength inversion process, which is the first in the drag-based editing task. It obviates the necessity for TTO and unlocks the generative capability of models. Therefore, LazyDrag naturally unifies precise geometric control with text guidance, enabling complex edits that were previously out of reach: opening the mouth of a dog and inpainting its interior, generating new objects like a "tennis ball", or for ambiguous drags, making context-aware changes like moving hands into pockets. Moreover, LazyDrag supports multi-round edits with simultaneous move and scale operations. Evaluated on DragBench, our method outperforms baselines in drag accuracy and perceptual quality, as validated by mean distances, VIEScore and user studies. LazyDrag not only sets new state-of-the-art performance, but also paves a new way to editing paradigms. Here is the project website.

## 1 Introduction

Drag-based editing in diffusion models remains fundamentally challenging. To preserve object identity during editing, prior methods often perform implicit point matching via attention. A common strategy, introduced by MasaCtrl (Cao et al., 2023), shares key and value tokens during attention. However, this strategy allocates more attention weights to spatially nearby regions instead of semantically related ones (Wang et al., 2025b; Feng et al., 2025), which leads to unstable and degrading edits. Rather than tackling this fundamental cause, as a compromise, many methods rely on test-time optimization (TTO) or weakened inversion strength. These compromises mask the mismatch and incur costs, including unreliable inpainting, suppressed text guidance, and distorted edits.

Instead of the compromise, we take a principled alternative: replace implicit attention-based matching with an explicit correspondence map and inject it directly into the generation process. With this reliable map, editing under full-strength inversion becomes stable without TTO, enabling faithful inpainting and text-guided generation. Beyond addressing the fundamental issue, the choice of network architecture remains crucial for editing. The recent transition from U-Nets (Rombach et al., 2022) to Multi-Modal Diffusion Transformers (MM-DiT) (Esser et al., 2024) provides an ideal foundation for this shift, because MM-DiTs offer tighter vision–text fusion, which improves inversion robustness and raises the ceiling for attention control. As shown by ColorCtrl (Yin et al., 2025b), this architecture supports stronger semantic consistency and controllability, allowing attention control to

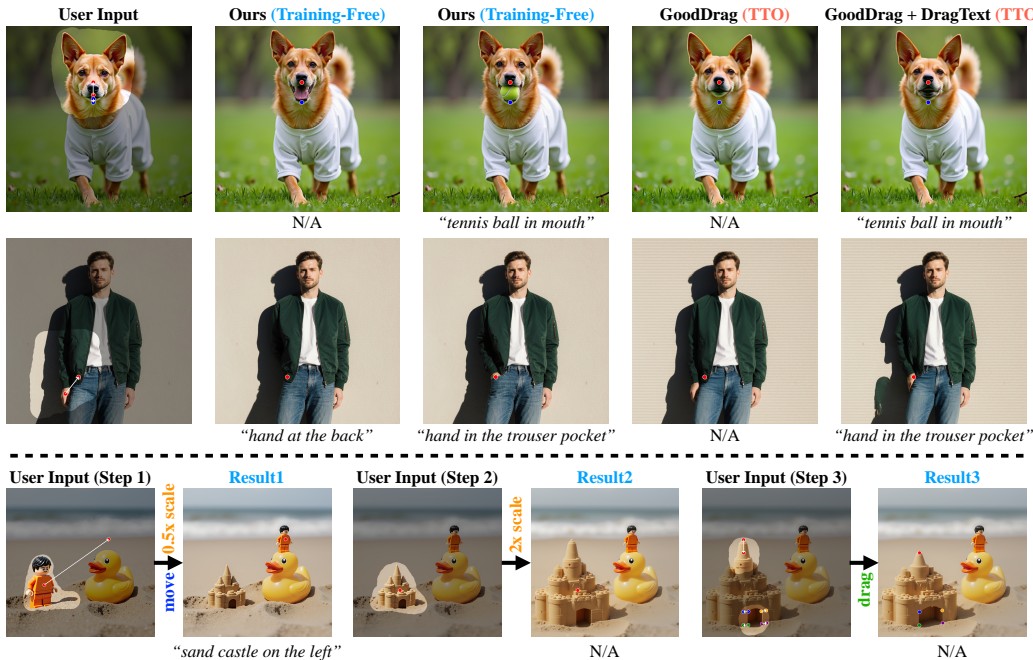

Figure 1: **(a) Top: Comparison between our method and two baselines.** The leftmost image shows the input image with multiple drag instructions, each indicated by a different color. The text below each result indicates the additional prompt used for generation. "N/A" means no additional prompt. TTO denotes test-time optimization, where the method requires fine-tuning per image and multi-step latent optimization per drag instruction. Notably, our method successfully opens the mouth of the dog and inpaints its interior. Furthermore, with prompt guidance, we can generate diverse results even under ambiguous drag inputs without fine-tuning. **(b) Bottom: Multi-round editing results using our approach.** Our method supports not only sequential drag operations but also simultaneous actions like movement and scaling, maintaining visual coherence throughout.

be applied across all single-stream attention (SS-Attn) layers without manual selection of specific layer indexes like that in U-Nets. We exploit these advantages by building our method on MM-DiTs.

Unlike in U-Nets, identity preservation in MM-DiTs is non-trivial. Simply sharing key and value tokens, as in DiTCtrl (Cai et al., 2025), does not reproduce the identity-preserving behavior achieved by MasaCtrl with U-Nets (Cao et al., 2023). Recently, CharaConsist (Wang et al., 2025b) showed that re-encoding and injecting semantically aligned tokens can preserve identity in MM-DiTs. However, its point matching relies on the average of attention similarity, which is fragile under full-strength inversion and often yields unsuitable edits. In contrast, drag instructions naturally define a field that maps handle points to target points, forming a deterministic correspondence map. We turn this explicit map into attention controls. This explicit correspondence–driven preservation resolves the root issue, stabilizes edits under full-strength inversion without TTO. As a result, it enhances inpainting and text guidance ability, delivering higher fidelity and controllability than prior methods.

In this work, we present **LazyDrag**, a training-free method that uses an explicit correspondence map to drive attention controls in MM-DiTs. By resolving the core instability of implicit attention mappings, LazyDrag stabilizes edits under full-strength inversion without TTO, unlocking the full generation ability. Concretely, (i) the drag instructions are converted into an explicit correspondence map, and (ii) identity and background are preserved using attention controls with the map. Together, these components deliver edits under full-strength inversion without TTO, retaining inpainting capability and enabling text-guided edits under ambiguous instructions. As shown in Fig. 1, this allows our method to execute complex edits where prior works fail: it can open the mouth of the dog and inpaint its interior, or even generate a "tennis ball" via text guidance, which is impossible for methods constrained by low inversion strength (see Fig. 2). Furthermore, it exhibits a deep understanding of scene context. For example, when dragging a hand using drag instructions alone, the ambiguity

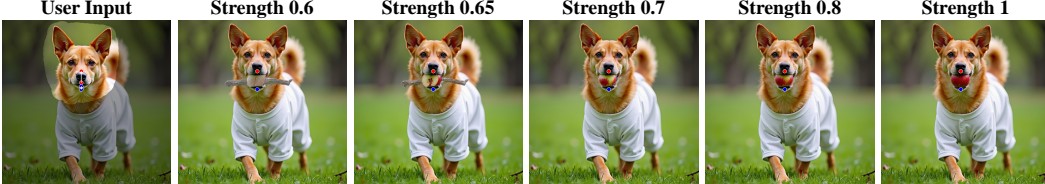

Figure 2: **Effect of inversion strength.** Examples of LazyDrag under different inversion strengths. The additional prompt is "a red apple in the mouth".

of the task, whether the hand should be placed behind a back or into a pocket, can be resolved through text guidance, allowing users to make precise and meaningful edits. Extensive experiments demonstrate that LazyDrag achieves **state-of-the-art** (SOTA) performance while requiring no test-time optimization. To the best of our knowledge, LazyDrag is the **first** drag-based editing method built with MM-DiTs and the **first** to adopt full-strength inversion across all sampling steps, which enables natural inpainting and precise text-guided control. Our contributions are threefold:

- We propose LazyDrag, the first to achieve full-strength inversion in drag-based editing with MM-DiTs. It is accomplished by an explicit correspondence-driven attention controls that eliminates the need for TTO and resolves the core instability of previous works.

- We resolve the ambiguity of drag instructions by coupling the explicit correspondence map with text guidance, enabling natural inpainting and semantically consistent edits.

- We resolve the ambiguity of drag instructions by coupling the explicit correspondence map with text guidance. This correspondence-driven method preserves identity and background, while enabling natural inpainting and semantically consistent modifications.

- Extensive experiments demonstrate that LazyDrag significantly outperforms all existing methods on Drag-Bench in both quantitative metrics and human preference.

## 2 RELATED WORK

**Text-to-image and video generation.** GAN-based models (Reed et al., 2016; Yu et al., 2023; Wang et al., 2023) have been largely replaced by diffusion models with U-Net backbones (Ho et al., 2020; Rombach et al., 2022) due to better fidelity and stability. However, U-Nets scale poorly, prompting a shift toward Diffusion Transformers (DiT) (Peebles & Xie, 2023). Among them, MM-DiT (Esser et al., 2024) has become the backbone of choice in recent state-of-the-art systems (Esser et al., 2024; AI, 2024; Labs, 2024; Yang et al., 2024; Kong et al., 2024; Liu et al., 2025a), including FLUX (Labs, 2025). We are the first to introduce a drag-based editing method within MM-DiTs.

**Text-based editing.** Training-free text-guided editing methods use pre-trained diffusion models without fine-tuning, offering strong flexibility. Prompt-to-Prompt (Hertz et al., 2023) edits attention maps for localized control, with extensions to images and videos (Wang et al., 2025a; Liu et al., 2024b; Cao et al., 2023; Rout et al., 2025; Xu et al., 2025; Ju et al., 2024; Yin et al., 2025a). Recent work explores attention control in MM-DiTs: DiTCtrl (Cai et al., 2025) for long video generation, ColorCtrl (Yin et al., 2025b) for light-consistent color edits, and CharaConsist (Wang et al., 2025b) for preserving character identity. Modern approaches such as Step1X-Edit (Liu et al., 2025b) and GPT-4o (OpenAI, 2025) have gained popularity due to their efficiency. However, all rely solely on text, which limits spatial precision. We instead introduce a more intuitive and controllable drag-based method.

**Drag-based editing.** Drag-based editing enables users to specify explicit spatial transformations by defining source and target points. Existing methods can be divided into two categories: those requiring test-time optimization (TTO), and those that do not. Most prior works fall into the former, beginning with DragGAN (Pan et al., 2023), and expanding to diffusion-based approaches (Shi et al., 2024b; Mou et al., 2024a;b; Liu et al., 2024a; Hou et al., 2024; Shin et al., 2024; Zhou et al., 2025; Ling et al., 2024; Zhang et al., 2025; Shi et al., 2024a). RegionDrag (Lu et al., 2024) extends

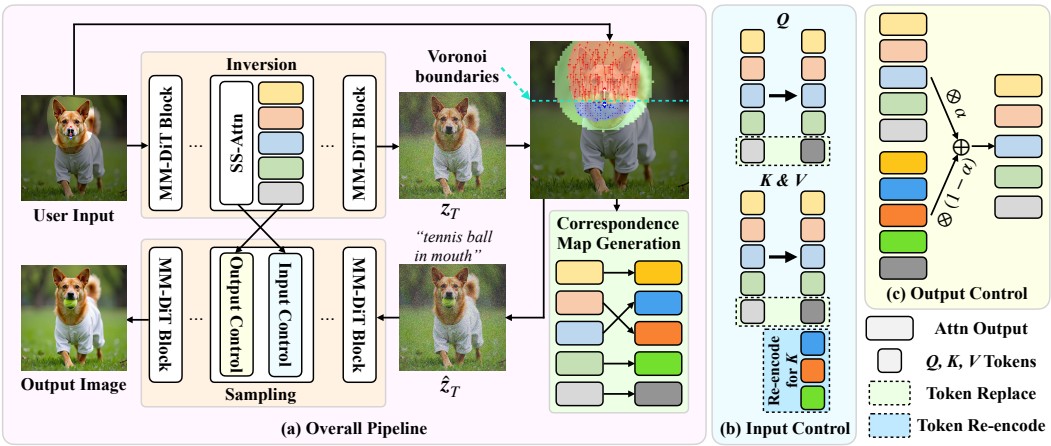

Figure 3: **Pipeline of LazyDrag.** (a) An input image is inverted to a latent code $z_T$. Our correspondence map generation then yields an updated latent $\hat{z}_T$, point matching map, and weights $\alpha$. Tokens cached during inversion are used to guide the sampling process for identity and background preservation. (b) In attention input control, a dual strategy is employed. For background regions (gray color), **Q**, **K**, and **V** tokens are replaced with their cached originals. For destination (red and blue colors) and transition regions (yellow color), the **K** and **V** tokens are concatenated with re-encoded (**K** only) source tokens retrieved via the map (c) Attention output refinement performs value blending of attention output. $\otimes$ and $\oplus$ denotes element-wise product and addition.

the interface to support region-level editing. Some methods (Jiang et al., 2025; Choi et al., 2025) incorporate textual prompts to improve semantic understanding, but still suffers from complex instructions. FastDrag (Zhao et al., 2024) is one of only two notable TTO-free methods, achieving faster inference but still falling short of the quality delivered by TTO-based methods. Inpaint4Drag (Lu & Han, 2025) is the other TTO-free method that build on an inpainting model rather than generative model with inversion. However, directly pasting a warped image to fill the edited region introduces strong unnatural warping artifacts. Also, its strong sensitivity to the input mask leads to frequent boundary artifacts and blurring, even with assistance from modern mask generators (*e.g.*, SAM (Kirillov et al., 2023)). Therefore, we adopt a widely used generative model approach with inversion, rather than an inpainting formulation. Additionally, all prior approaches with inversion rely on low inversion strength, which degrades inpainting quality and limits semantic generation. In contrast, we introduce the first drag-based method for MM-DiTs that leverages full-strength inversion and text-guided attention mechanisms, achieving SOTA performance without any per-image tuning.

## 3 METHOD

Our goal is to achieve identity-preserving edits with precise drag control, text guidance, and natural inpainting. To this end, we introduce **LazyDrag**, a training-free method built with MM-DiTs under full-strength inversion property. Our approach replaces the fragile, implicit point matching of prior work with a robust, explicit correspondence map derived from user input during attention control, stabilizing the inversion process without test-time optimization. We first review foundational concepts in Sec. 3.1. Then detail our two-stage approach: first, how to generate the explicit correspondence map from drag instructions (Sec. 3.2), and second, how this map drives a novel two-part attention control for identity and background preservation (Sec. 3.3). Fig. 3 shows the pipeline.

### 3.1 PRELIMINARIES

LazyDrag builds upon insights from training-free drag-based editing methods in U-Nets (Sec. 3.1.1) and identity preservation in MM-DiTs (Sec. 3.1.2), addressing core limitations of both (Sec. 3.1.3).

### 3.1.1 TRAINING-FREE DRAG EDITING IN U-NETS: FASTDRAG.

FastDrag (Zhao et al., 2024) is the first training-free method for drag-based editing, with U-Net models. It has two parts: (1) it computes a displacement field from drag instructions to create an initial latent $\hat{z}_T$, filling exposed regions via interpolation, and (2) it applies a MasaCtrl-like (Cao et al., 2023) key and value token replacement during self-attention to preserve object identity. However, beyond the implicit locality bias of self-attention, a central trade-off arises: we want handle points to reach their targets while surrounding regions inpaint naturally. Yet after latent initialization, the cue specific to handles is lost, and all moved points are treated uniformly. Forcing exact positional accuracy yields warp artifacts, whereas enforcing naturalness reduces positioning accuracy. Thus, editing accuracy and visual fidelity are in inherent tension. Moreover, its fusion of multiple instructions is brittle: when drags are antagonistic (for example, opening a mouth by moving the upper lip upward and the lower lip downward), averaging the displacements cancels motion near the seam and the mouth fails to open. Moreover, the interpolation used to fill newly exposed regions further replicates nearby textures, producing repeated artifacts in large uncovered areas, as shown in Fig. 11.

### 3.1.2 IDENTITY PRESERVATION IN MM-DITS: CHARACONSIST.

In parallel, CharaConsist (Wang et al., 2025b) introduces identity preservation in MM-DiTs, though it is not an editing method. To enforce identity preservation, it controls attention by concatenating corresponding source tokens into the key (re-encoded) and value tokens and by blending attention outputs. However, its point matching mechanism is critically flawed: it relies on attention similarity to identify matching points between images, a process that is computationally expensive (requiring additional denoising steps) and inherently unstable. Under full-strength inversion, even minor mismatches in the correspondence map can lead to significant visual artifacts, as proved in Tab. 3.

### 3.1.3 LAZYDRAG: BRIDGING THE GAP.

Naively extending FastDrag from U-Nets to MM-DiTs and combining it with the attention control methods of CharaConsist exposes and amplifies their respective weaknesses, yielding unusable results, as shown in Fig. 6 and Tab. 3. LazyDrag resolves these weaknesses with a unified solution: an explicit correspondence map derived from drag instructions. This map provides stable, precise attention control throughout the generation process, enabling high-quality, accurate edits while avoiding the pitfalls of attention-similarity matching and the trade-offs inherent in FastDrag.

## 3.2 GENERATING THE EXPLICIT CORRESPONDENCE MAP

We first compute an **explicit correspondence map** from the user drag instructions and the inverted source latent noise $z_T$. The map comprises a matching point function $\mathcal{M}$ and a weight function $\mathcal{A}$, which provides explicit guidance. Guided by this map, we generate the initial latent noise $\hat{z}_T$.

**Displacement field calculation via winner-takes-all (WTA).** Let $\Omega$ denote the latent grid, and let $\mathcal{P} = \{p_j\}_{j=1}^m \subset \Omega$ be the editable regions (the bright area in Fig. 3), sampled as feature points. Let the drag instructions be $\mathcal{D} = \{(s_i, e_i)\}_{i=1}^k$, where $s_i$ and $e_i$ are the handle and target points of the $i$-th instruction. We illustrate two modes for computing the displacement field. In **drag** mode, we adopt the elasticity-based per-instruction displacement $v_j^i$ for each $p_j$ under the $i$-th instruction as in Zhao et al. (2024); in **move** mode, we use standard translation and scaling. To avoid failures of averaging under opposing drags, we use a robust **winner-takes-all** (Aurenhammer, 1991) fusion: each $p_j$ is uniquely assigned to its nearest handle, inducing a Voronoi partition (Aurenhammer, 1991). The final displacement $v_j$ and weight $\alpha_j$ are determined solely by the winning instruction.

$$\alpha_j^i = \begin{cases} \|p_j - s_i\|_2^{-1}, & p_j \neq s_i, \\ \infty, & \text{otherwise,} \end{cases}$$

$$v_j = v_j^{i^\star}, \quad \alpha_j = \alpha_j^{i^\star}, \quad \text{where} \quad i^\star = \arg\max_i \alpha_j^i. \tag{1}$$

Here, $\|\cdot\|_2$ denotes the Euclidean $L_2$-norm distance. Thus, $\mathcal{V} = \{v_j\}_{j=1}^m$ is defined as the displacement field. This approach preserves the full magnitude of opposing drags, enabling complex edits like opening a mouth, which is impossible with simple averaging. Details are in Appendix A.2.

**Initial latent construction and map formalization (Latent Init).** With the displacement field $\mathcal{V}$ established, we construct the initial latent $\hat{z}_T$. This process defines our explicit deterministic correspondence map $(\mathcal{M}, \mathcal{A})$ and partitions the latent grid into distinct regions for targeted control. First, we define the set of discrete destination coordinates $\mathcal{P}^\star = \{\Pi(\boldsymbol{p}_j + \boldsymbol{v}_j) \mid \boldsymbol{p}_j \in \mathcal{P}\}$, where $\Pi(\cdot)$ projects to the grid. By resolving collisions where multiple source points map to a single destination $\boldsymbol{x} \in \mathcal{P}^\star$ (using winner-takes-all), we get the winner index $j^\star(\boldsymbol{x}) = \arg\max_{j:\,\Pi(\boldsymbol{p}_j + \boldsymbol{v}_j) = \boldsymbol{x}} \alpha_j$ and formalize our correspondence map: **Matching point map**, $\mathcal{M}(\boldsymbol{x}) = \boldsymbol{p}_{j^\star(\boldsymbol{x})}$. **Matching weight map**, $\mathcal{A}(\boldsymbol{x}) = \min(1, \alpha_{j^\star(\boldsymbol{x})})$. Next, we partition the latent space $\Omega$ into four disjoint sets based on the geometry of the warp. These sets correspond directly to the colored regions in Fig. 3 (a): *Background* $\mathcal{R}^{\mathrm{bg}}$ (gray) that must remain unchanged, *Destinations* $\mathcal{R}^{\mathrm{dst}}$ (red and blue, *a.k.a.*, $\mathcal{P}^\star$) where moved content is rendered with identity preserved, *Inpainting* $\mathcal{R}^{\mathrm{inp}}$ (yellow) initialized from noise, and *Transition* $\mathcal{R}^{\mathrm{trans}}$ (green) that blends boundaries smoothly. With these regions clearly defined, the updated latent $\hat{z}_T$ is constructed by applying a specific replacement rule to each region:

$$\hat{\boldsymbol{z}}_T(\boldsymbol{x}) = \begin{cases} \boldsymbol{z}_T(\mathcal{M}(\boldsymbol{x})), & \text{if } \boldsymbol{x} \in \mathcal{R}^{\mathrm{dst}}, \\ \boldsymbol{\epsilon}(\boldsymbol{x}), & \text{if } \boldsymbol{x} \in \mathcal{R}^{\mathrm{inp}}, \\ \boldsymbol{z}_T(\boldsymbol{x}), & \text{if } \boldsymbol{x} \in \mathcal{R}^{\mathrm{bg}} \cup \mathcal{R}^{\mathrm{trans}}, \end{cases} \tag{2}$$

where $\boldsymbol{\epsilon} \sim \mathcal{N}(0, \mathbf{I})$. Crucially, replacing the BNNI interpolation used in FastDrag with **Gaussian noise** in $\mathcal{R}^{\mathrm{inp}}$ is essential. Unlike the uniform noise compared in Zhao et al. (2024), this approach aligns with the diffusion prior, prevents repetitive artifacts as shown in Fig. 11, and enables the ability of high-fidelity, text-guided inpainting discussed in the introduction.

### 3.3 Correspondence-Driven Preservation

Having established the explicit correspondence map, we now detail a two-part mechanism operating at the input (Sec. 3.3.1) and output (Sec. 3.3.2) of the attention calculation in single-stream attention layers only (Yin et al., 2025b; Deng et al., 2025). Using this map, the mechanism provides fine-grained control that preserves identity and background, ensuring robust full-strength inversion.

#### 3.3.1 Attention Input Control via Token Replacement and Concatenation

To preserve the background and identity, the first part modifies the attention inputs of different regions. Let $(\mathbf{Q}_{\boldsymbol{x}}, \mathbf{K}_{\boldsymbol{x}}, \mathbf{V}_{\boldsymbol{x}})$ denote the current attention tokens at position $\boldsymbol{x}$ in a given layer and step, and $(\overline{\mathbf{Q}}_{\boldsymbol{x}}, \overline{\mathbf{K}}_{\boldsymbol{x}}, \overline{\mathbf{V}}_{\boldsymbol{x}})$ the tokens cached without positional encoding during the previous inversion process. Let $\mathrm{RoPE}_{\boldsymbol{x}}(\cdot)$ re-encode tokens with the rotary embedding at position $\boldsymbol{x}$ (Su et al., 2024).

**Background preservation via replacement (BG Pres.).** For the background region $\mathcal{R}^{\mathrm{bg}}$, the purpose of absolute untouched is achieved by hard-replacing the attention tokens with their cached originals at every step and every single-stream layer, similar to ColorCtrl (Yin et al., 2025b):

$$(\mathbf{Q}_{\boldsymbol{x}}, \mathbf{K}_{\boldsymbol{x}}, \mathbf{V}_{\boldsymbol{x}}) \leftarrow (\mathrm{RoPE}_{\boldsymbol{x}}(\overline{\mathbf{Q}}_{\boldsymbol{x}}), \mathrm{RoPE}_{\boldsymbol{x}}(\overline{\mathbf{K}}_{\boldsymbol{x}}), \overline{\mathbf{V}}_{\boldsymbol{x}}), \quad \forall \boldsymbol{x} \in \mathcal{R}^{\mathrm{bg}}. \tag{3}$$

**Identity preservation via concatenation (ID Pres.).** For the destination and transition regions ($\mathcal{R}^{\mathrm{dst}} \cup \mathcal{R}^{\mathrm{trans}}$), where identity must be preserved while allowing for coherent adaptation, we use token concatenation. Define a unified source point map, $\tilde{\mathcal{M}}(\boldsymbol{x})$, which selects correspondence sources:

$$\tilde{\mathcal{M}}(\boldsymbol{x}) = \begin{cases} \mathcal{M}(\boldsymbol{x}), & \text{if } \boldsymbol{x} \in \mathcal{R}^{\mathrm{dst}}, \\ \boldsymbol{x}, & \text{if } \boldsymbol{x} \in \mathcal{R}^{\mathrm{trans}}. \end{cases} \tag{4}$$

For any position $\boldsymbol{x} \in \mathcal{R}^{\mathrm{dst}} \cup \mathcal{R}^{\mathrm{trans}}$, we form an augmented key $\mathbf{K}'_{\boldsymbol{x}}$ and value $\mathbf{V}'_{\boldsymbol{x}}$ by concatenating the cached tokens from its designated source $\tilde{\mathcal{M}}(\boldsymbol{x})$:

$$\mathbf{K}'_{\boldsymbol{x}} = \mathrm{concat}\left(\mathbf{K}_{\boldsymbol{x}}, \mathrm{RoPE}_{\boldsymbol{x}}(\overline{\mathbf{K}}_{\tilde{\mathcal{M}}(\boldsymbol{x})})\right), \tag{5}$$

$$\mathbf{V}'_{\boldsymbol{x}} = \mathrm{concat}\left(\mathbf{V}_{\boldsymbol{x}}, \overline{\mathbf{V}}_{\tilde{\mathcal{M}}(\boldsymbol{x})}\right). \tag{6}$$

This provides a strong, correspondence-driven signal to the attention calculation, robustly preserving identity while allowing for smooth blending at the boundaries.

### 3.3.2 Attention Output Refinement via Gated Merging (Attn Refine)

The second part refines the attention output so that it cooperates with the above token concatenation (following Wang et al. (2025b)), improving visual quality and emphasizing the importance of handle points over others. Let $\mathbf{y}_{\boldsymbol{x}}$ be the attention output at $\boldsymbol{x}$ and $\overline{\mathbf{y}}_{\boldsymbol{x}}$ be the cached output. For $\boldsymbol{x} \in \mathcal{R}^{\mathrm{dst}}$,

$$\mathbf{y}_{\boldsymbol{x}} \leftarrow \left(1 - \gamma_{\boldsymbol{x},t}\right) \mathbf{y}_{\boldsymbol{x}} + \gamma_{\boldsymbol{x},t}\, \overline{\mathbf{y}}_{\mathcal{M}(\boldsymbol{x})}, \tag{7}$$

where the blending factor $\gamma_{\boldsymbol{x},t}$ is gated by our pre-computed matching weight from the map $\mathcal{A}$:

$$\gamma_{\boldsymbol{x},t} = h_t \cdot \mathcal{A}(\boldsymbol{x}), \tag{8}$$

where $t$ indexes the timestep and $h_t \in [0,1]$ is a factor that decays over time. This correspondence-driven *gated merge* eliminating the extra denoising steps required by CharaConsist, and addressing the instability of attention-similarity matching and scaling under full-strength inversion. By making the weight strongest at the handle points (where $\mathcal{A}(\boldsymbol{x})$ is maximal), it ensures precise control where it matters most, removing the need for multi-step latent optimization in previous methods (Zhang et al., 2025; Shi et al., 2024b), while allowing for natural relaxation in surrounding regions.

## 4 Experiments

### 4.1 Setup

**Baselines.**  We compare against eight baselines: DragDiffusion (Shi et al., 2024b), DragNoise (Liu et al., 2024a), FreeDrag (Ling et al., 2024), DiffEditor (Mou et al., 2024a), GoodDrag (Zhang et al., 2025), DragText (Choi et al., 2025)[1], FastDrag (Zhao et al., 2024), and Inpaint4Drag (Lu & Han, 2025). Notably, all baselines are U-Net–based, whereas ours is the first MM-DiT-based method.

**Implementation details.**  Unless otherwise noted, all baselines are run with their official implementations and default hyperparameters. For Inpaint4Drag (Lu & Han, 2025), we adopt the refined masks and point pairs provided by the authors at inference, and replace distilled models with original models. Our method is built on FLUX.1 Krea-dev (Labs, 2025), adopting the inversion method of UniEdit-Flow (Jiao et al., 2025) while replacing the editing strategy with our approach. Following CharaConsist (Wang et al., 2025b), we activate ID Pres. and Attn Refine (Sec. 3.3) for the first 40 denoising steps, referring to the last activate timestep as the **activation timestep**. For a fair comparison, the number of denoising steps is fixed to 50 for all methods. More details are in Appendix A.1.

**Benchmark and evaluation protocol.**  We evaluate on DragBench (Shi et al., 2024b), which contains 205 images with 349 handle and target point pairs. Our primary accuracy metric is **MD** (mean distance) (Pan et al., 2023). Although IF (image fidelity) (Kawar et al., 2023), typically computed with LPIPS (Zhang et al., 2018), is widely used, we *do not* report IF. Previous work (Choi et al., 2025; Lu et al., 2024) shows that successful drag edits necessarily change the image, often increasing LPIPS, whereas an unchanged image trivially attains the best score. Hence, IF can be misleading for drag editing. To obtain a complementary, perceptually grounded view, we adopt the VIEScore (Ku et al., 2024) metrics from GEdit-Bench (Liu et al., 2025b): **SC** (Semantic Consistency): whether the intended edit has been achieved. **PQ** (Perceptual Quality): the naturalness of the result and absence of artifacts. **O** (Overall): the overall performance defined in Liu et al. (2025b). In our setting, the "intended edit" is specified by the dragging instruction rather than a natural-language instruction, but the scoring criteria remain unchanged. Each score ranges from 0 to 10 (higher is better) and is produced by the state-of-the-art MLLM evaluator, GPT-4o[2] (Hurst et al., 2024). To mitigate stochasticity in evaluation, we run every evaluation metrics three times and report both the mean and standard deviation. We additionally report a binary **TTO-Req** (Test-Time Optimization Required) flag indicating whether a method requires per-edit test-time optimization (*e.g.*, LoRA fine-tuning or multi-step latent optimization) during inference. More evaluation details are in Appendix A.3.

---

[1]Since DragText is a plug-and-play method, we evaluate it in conjunction with best-performing GoodDrag.
[2]API access as of August 2025

Table 1: Quantitative results compared with baselines on Drag-Bench.

| Method | TTO-Req | MD ↓ | SC ↑ | PQ ↑ | O ↑ |
|--------|---------|------|------|------|-----|
| DragNoise (Liu et al., 2024a) | ✓ | 37.87 ± 0.23 | 7.793 ± 0.04 | 8.058 ± 0.01 | 7.704 ± 0.01 |
| DragDiffusion (Shi et al., 2024b) | ✓ | 34.84 ± 0.30 | 7.905 ± 0.01 | 8.325 ± 0.02 | 7.798 ± 0.03 |
| FreeDrag (Ling et al., 2024) | ✓ | 34.09 ± 0.60 | 7.928 ± 0.02 | 8.281 ± 0.03 | 7.816 ± 0.02 |
| DiffEditor (Mou et al., 2024a) | ✓ | 26.95 ± 0.24 | 7.603 ± 0.01 | 8.266 ± 0.01 | 7.715 ± 0.01 |
| GoodDrag (Zhang et al., 2025) | ✓ | 22.17 ± 0.16 | 7.834 ± 0.03 | 8.318 ± 0.01 | 7.795 ± 0.01 |
| DragText (Choi et al., 2025) | ✓ | 21.51 ± 0.21 | 7.992 ± 0.02 | 8.227 ± 0.01 | 7.886 ± 0.01 |
| FastDrag (Zhao et al., 2024) | ✗ | 31.84 ± 0.96 | 7.935 ± 0.09 | 8.278 ± 0.01 | 7.904 ± 0.06 |
| Inpaint4Drag (Lu & Han, 2025) | ✗ | 23.68 ± 0.05 | 7.802 ± 0.06 | 7.961 ± 0.04 | 7.615 ± 0.06 |
| Ours | ✗ | **21.49** ± 0.04 | **8.205** ± 0.03 | **8.395** ± 0.03 | **8.210** ± 0.03 |

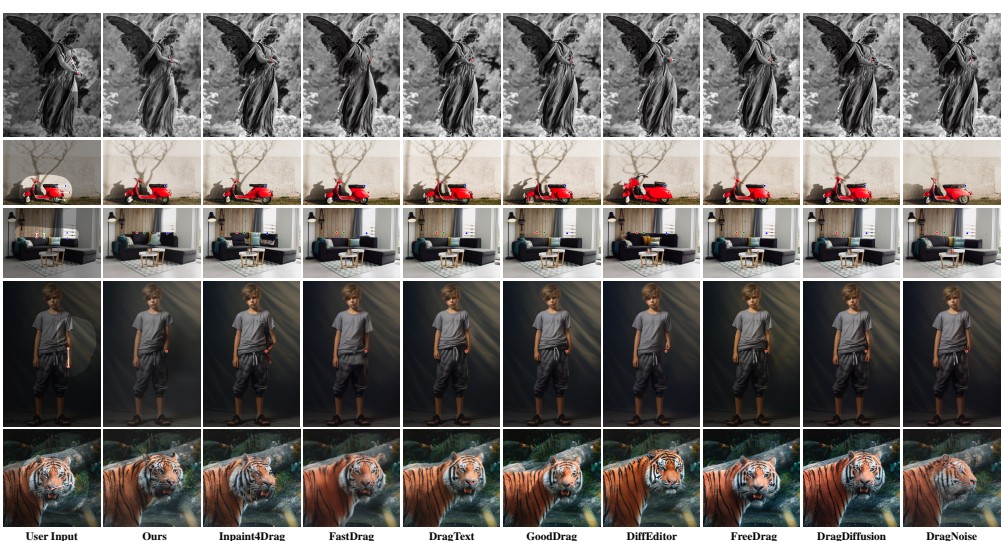

Figure 4: Qualitative results compared with baselines on Drag-Bench. ***Best viewed with zoom-in.***

## 4.2 QUANTITATIVE EVALUATION

Tab. 1 presents the benchmark results on DragBench. Despite not requiring LoRA fine-tuning or multi-step latent optimization for each image and drag operation, our method consistently outperforms existing approaches in all metrics, especially in terms of drag accuracy and the perceptual quality of the generated images. Notably, our approach achieves SOTA performance out-of-the-box, without the need for test-time optimization, making it both efficient and effective. Specifically, Inpaint4Drag (Lu & Han, 2025) often produces boundary artifacts and color shifts between edited and unedited regions. Consequently, the LLM evaluator assigns lower scores under its over-editing rule. This indicates that, even with additional optimization of masks and point pairs, mask sensitivity of inpainting models degrades results. By contrast, our full-strength inversion method with attention controls attains strong performance while being more robust to the choice of masks and point pairs.

## 4.3 QUALITATIVE EVALUATION

Fig. 4 qualitatively demonstrates the superiority of our method over existing baselines. In the first example, only our method correctly lift the arm with background maintained, while others introduce artifacts, such as distorted hands (*e.g.*, DragText (Choi et al., 2025)) or unintended background changes (*e.g.*, DragNoise (Liu et al., 2024a)). In the second example, most baselines fail to preserve the front structure of the vehicle, whereas our approach maintains it faithfully while applying the desired transformation. Specifically, Inpaint4Drag (Lu & Han, 2025) generates artifacts in the background. The third case shows that only our method successfully modifies the sofa geometry while preserving the integrity of pillows. In the fourth example, our approach correctly interprets hand proximity as intent to insert it into the pocket, while other baselines introducing artifacts. Finally, in

Table 2: User study on Drag-Bench.

| Method | Preference (%) |
|---|---|
| DragNoise (Liu et al., 2024a) | $6.64 \pm 8$ |
| DragDiffusion (Shi et al., 2024b) | $6.25 \pm 8$ |
| FreeDrag (Ling et al., 2024) | $6.64 \pm 7$ |
| DiffEditor (Mou et al., 2024a) | $2.34 \pm 4$ |
| GoodDrag (Zhang et al., 2025) | $5.86 \pm 5$ |
| DragText (Choi et al., 2025) | $2.73 \pm 4$ |
| FastDrag (Zhao et al., 2024) | $2.34 \pm 5$ |
| Inpaint4Drag (Lu & Han, 2025) | $3.13 \pm 4$ |
| Ours | $\mathbf{63.67} \pm 16$ |

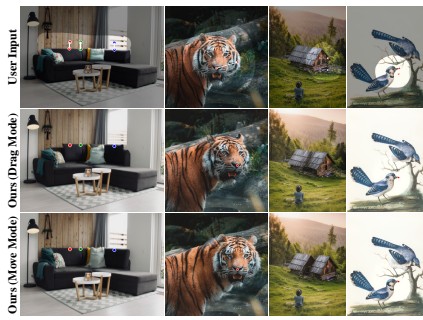

Figure 5: Comparison between drag and move mode on Drag-Bench.

the fifth example, only our approach and DragText successfully rotates the head of the tiger to the right without compromising overall image quality. These results are consistent with our quantitative evaluations and highlight the robustness and generality of our method, even without per image tuning or per instruction multi-step latent optimization. More results are shown in Appendix B.

## 4.4 USER STUDY

A total of 32 expert participants evaluated comparisons between methods on 32 cases randomly sampled from DragBench. For each comparison, method order positions were randomized and method identities were anonymized. Participants selected the preferred result according to predefined criteria (edit success, naturalness, and background preservation). Overall, LazyDrag was preferred in 61.88% of comparisons, outperforming all baselines (Tab. 2). More details are in Appendix A.4.

## 4.5 COMPARISON BETWEEN DRAG AND MOVE MODES

We evaluate LazyDrag with both drag and move modes on Drag-Bench, with qualitative results shown in Fig. 5. The move mode tends to better preserve identity, as seen in the last two cases, rather than performing edits involving rotation or extension, as in the second and third examples. In contrast, the drag mode enables natural geometric transformations, including 3D rotations and extensions, albeit with a slight degradation in detail texture preservation. Both of two modes can generate reasonable results. These findings highlight the flexibility of our explicit correspondence map when paired with our correspondence-driven preservation strategy. Future work may explore more matching strategies, such as 2D rotation, to further enhance diversity and controllability.

## 4.6 ABLATION STUDY

**Effect of each component.** We conduct an ablation study in which components are progressively removed from the full method. To keep functionality comparable when a component is absent, we adopt controlled replacements: (i) Without WTA and Latent Init (Sec. 3.2) we revert to latent warpage optimization of FastDrag (Zhao et al., 2024) as the latent initialization. (ii) Without ID Pres. and Attn Refine (Sec. 3.3) we switch to the attention-similarity matching and scaling introduced in CharaConsist (Wang et al., 2025b). Fig. 6 and Tab. 3 report benchmark results on Drag-Bench. Removing WTA and Latent Init increases **MD** and slightly reduces **PQ** and **O**, indicating that our initialization with the winner-takes-all fusion strategy and random initialization for inpainting regions suppresses repetitive artifacts and improves inpainting quality as proven in the figure. Further disabling background preservation causes additional drops in **SC** and **O** due to color shifting and artifacts in the background. Finally, replacing our correspondence-driven preservation with attention-similarity control leads to a sharp degradation, highlighting the sensitivity of full-strength inversion to mismatched attention alignment. The full method achieves the best performance.

**Effect of activation timesteps.** We conduct an ablation study on the effect of activation timesteps by varying the activation timestep to 20, 40, and 50, as shown in Fig. 7 and Tab. 4. From the results, we observe that increasing the number of the activation timestep leads to more accurate destination points for dragging, though it may introduce more warping artifacts. Conversely, reducing the acti-

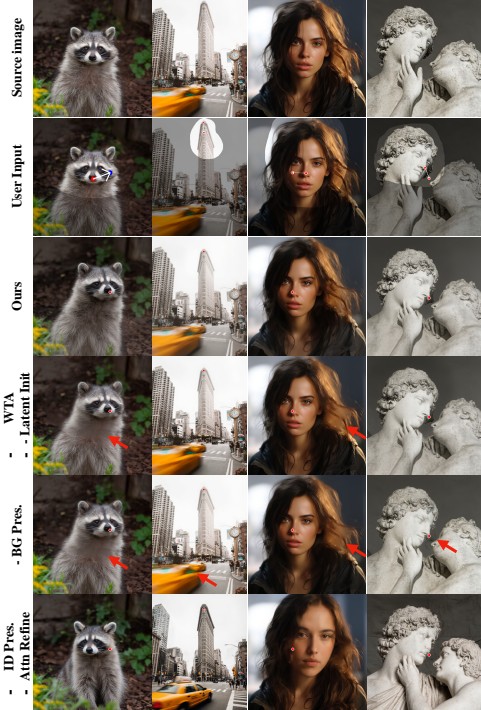

Figure 6: **Qualitative cumulative ablation on Drag-Bench.** Rows remove one component relative to the row above. When WTA and Latent Init are removed we use latent init in FastDrag. When ID Pres. and Attn Refine are removed we switch to CharaConsist attention-similarity control.

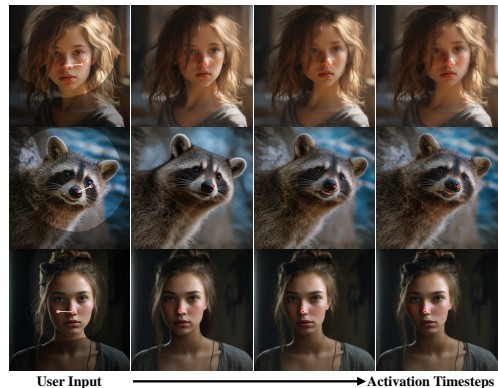

Figure 7: **Qualitative ablation of activation timesteps on Drag-Bench.** From left to right, the activation timestep is increased.

Table 3: Quantitative cumulative ablation on Drag-Bench under the same setting as Fig. 6

| Method | MD ↓ | SC ↑ | PQ ↑ | O ↑ |
|---|---|---|---|---|
| Ours | **21.49** ± 0.04 | **8.205** ± 0.03 | **8.395** ± 0.03 | **8.210** ± 0.03 |
| - WTA - Latent Init | 23.69 ± 0.16 | 8.129 ± 0.03 | 8.060 ± 0.05 | 7.938 ± 0.01 |
| - BG Pres. | 24.73 ± 0.08 | 7.998 ± 0.04 | 8.043 ± 0.01 | 7.863 ± 0.03 |
| - ID Pres. - Attn Refine | 56.49 ± 0.49 | 5.307 ± 0.08 | 7.944 ± 0.02 | 5.953 ± 0.06 |

Table 4: Quantitative ablation of activation timesteps on Drag-Bench.

| Method | MD ↓ | SC ↑ | PQ ↑ | O ↑ |
|---|---|---|---|---|
| Ours (40 as activation timestep) | **21.49** ± 0.04 | 8.205 ± 0.03 | 8.395 ± 0.03 | **8.210** ± 0.03 |
| + 20 as activation timestep | 34.23 ± 0.29 | 7.036 ± 0.03 | **8.788** ± 0.01 | 7.605 ± 0.02 |
| + 50 as activation timestep | 21.81 ± 0.26 | **8.298** ± 0.03 | 8.072 ± 0.01 | 8.087 ± 0.03 |

vation timestep results in more natural outputs, but may cause slight variations in identity or motion. More results are given in Appendix B.5. For benchmark evaluations, we use 40 as a balanced value.

**Additional results.** Appendix B presents additional evaluations on Drag-Bench, ablation studies with U-Nets, effects of text guidance, runtime analysis, and limitations.

## 5 CONCLUSION

We presented **LazyDrag**, the first training-free method for drag-based editing with MM-DiTs under full-strength inversion. We begin by identifying the fundamental cause of instability in drag-based editing: the unreliability of implicit attention-based point matching. This diagnosis explains why prior methods adopted compromises such as test-time optimization or weakened inversion strength, which suppress text guidance, harm inpainting, and limit generative ability. Our approach directly solves this core issue by replacing fragile implicit point matching with an explicit correspondence map that drives attention controls during generation. This correspondence-driven preservation enables robust edits under full-strength inversion without TTO. As a result, LazyDrag preserves identity and background, supports faithful inpainting, and leverages text guidance to resolve ambiguity in drag instructions. Extensive experiments show that LazyDrag achieves state-of-the-art performance, unifying precise control with text guidance to execute complex semantic edits. By revealing that the perceived stability–quality compromise is an artifact of flawed point matching, LazyDrag establishes a more powerful and principled foundation for future research and marks a concrete step toward intuitive, AI-native creative workflows and more sophisticated generative control.

ETHICS STATEMENT

The development of advanced image editing technologies inevitably raises important ethical concerns. Although our method enhances editing precision through text and drag-based controls, it also introduces potential risks, including the creation of misleading or harmful visual content. To address this, we emphasize the importance of using such tools responsibly, with clear attention to transparency and user consent in practical deployments. In addition, the underlying pre-trained models may encode and reproduce societal biases, which could influence the outputs in unintended ways. We view this as an open research challenge and encourage future work aimed at bias detection and mitigation. All human evaluation participants were fully informed of the purpose of the study and provided consent before participation.

REPRODUCIBILITY STATEMENT

We have made every effort to ensure the reproducibility of LazyDrag. Detailed descriptions of the inference procedure and evaluation settings are provided in Sec. 3, Sec. 4.1 and Appendix A. All source code will be released to the public upon acceptance of this paper, enabling researchers to fully replicate and build upon our results.

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

# A   IMPLEMENTATION DETAILS

## A.1   INFERENCE SETTINGS

For all baselines, we use their official code with default hyperparameters for inference. The number of denoising steps is set to 50, and classifier-free guidance (CFG) (Ho & Salimans, 2021) is set to 1. All images on Drag-Bench are generated at their original resolution, while other images are generated at $1024 \times 1024$. All generations are performed on a single NVIDIA H800 GPU.

EasyDrag (Hou et al., 2024) and CLIPDrag (Jiang et al., 2025) are excluded from comparison because their released implementations either fail to execute reliably or do not reproduce the results reported in the papers. DragGAN (Pan et al., 2023) is also excluded due to its inferior performance and slower processing speed compared to diffusion-based methods, as demonstrated in GoodDrag (Zhang et al., 2025).

For Inpaint4Drag, we remove the LCM (Luo et al., 2023) LoRA and fix the number of denoising steps to 50. We also replace the distilled VAE (Kingma & Welling, 2013) with the original VAE to improve reconstruction and generation quality. These settings are chosen to obtain the strongest editing performance rather than to optimize for speed.

For our inversion process, we adopt the official inversion method of UniEdit-Flow (Jiao et al., 2025) but replace the editing component with our proposed strategy. We apply our correspondence-driven preservation (Sec. 3.3) only to the single-stream attention layers in FLUX.1 Krea-dev (Labs, 2025). Since additional manipulation in dual-stream attention layers does not lead to noticeable improvements (Deng et al., 2025; Yin et al., 2025b; Wang et al., 2025a), we adopt a more efficient and concise design by limiting modifications to single-stream layers only.

## A.2   IMPLEMENTATION DETAILS OF DISPLACEMENT FIELD CALCULATION

**Per-instruction displacement.**   Following the principles of elasticity (Naylor, 1969; Zhao et al., 2024), the influence of an external force decays inversely with distance from the force origin, and the direction of the induced displacement aligns with the direction of the applied force. We represent each drag instruction $d_i$ as a vector from source $s_i$ to target $e_i$. For $p_j \in \mathcal{P}$, we write

$$v_j^i = \lambda_j^i \, d_i, \tag{9}$$

where $\lambda_j^i$ is a stretch factor. Using a reference circle $O$ that circumscribes the bounding rectangle of $\mathcal{P}$, extend the ray $s_i \to p_j$ to intersect $O$ at $q_j^i$. Enforcing parallelism between $v_j^i$ and $d_i$ yields

$$\lambda_j^i = \frac{\|v_j^i\|_2}{\|d_i\|_2} = \frac{\|p_j - p_j^i\|_2}{\|s_i - e_i\|_2} = \frac{\|p_j - q_j^i\|_2}{\|s_i - q_j^i\|_2}. \tag{10}$$

**Winner-takes-all blending.**   Weighted averaging multiple instruction can fail when different drags point in opposite directions. We therefore assign each $p_j$ to its nearest handle $s_i$ (a Voronoi partition (Aurenhammer, 1991)) as illustrated in Fig. 3 (a), where the red and blue regions correspond to two drag instructions, with weights

$$\alpha_j^i = \begin{cases} \|p_j - s_i\|_2^{-1}, & p_j \neq s_i, \\ \infty, & \text{otherwise.} \end{cases} \tag{11}$$

The final displacement is determined by the winning instruction $i^\star = \arg\max_i \alpha_j^i$:

$$v_j = v_j^{i^\star} = \lambda_j^{i^\star} \, d_{i^\star}. \tag{12}$$

This yields sharper spatial separation and avoids interference between opposing drags.

**Unified move/scale model.**   For axis-aligned resizing, we introduce a scaling vector $r \in \mathbb{R}^2$ to form a unified model:

$$v_j = \lambda_j^{i^\star} \, d_{i^\star} + (r - 1) \otimes (p_j - s_{i^\star}), \tag{13}$$

where $\otimes$ denotes element-wise product. For a move-and-scale operation, we set $\lambda_j^{i^\star} = \alpha_j^{i^\star} = 1$.

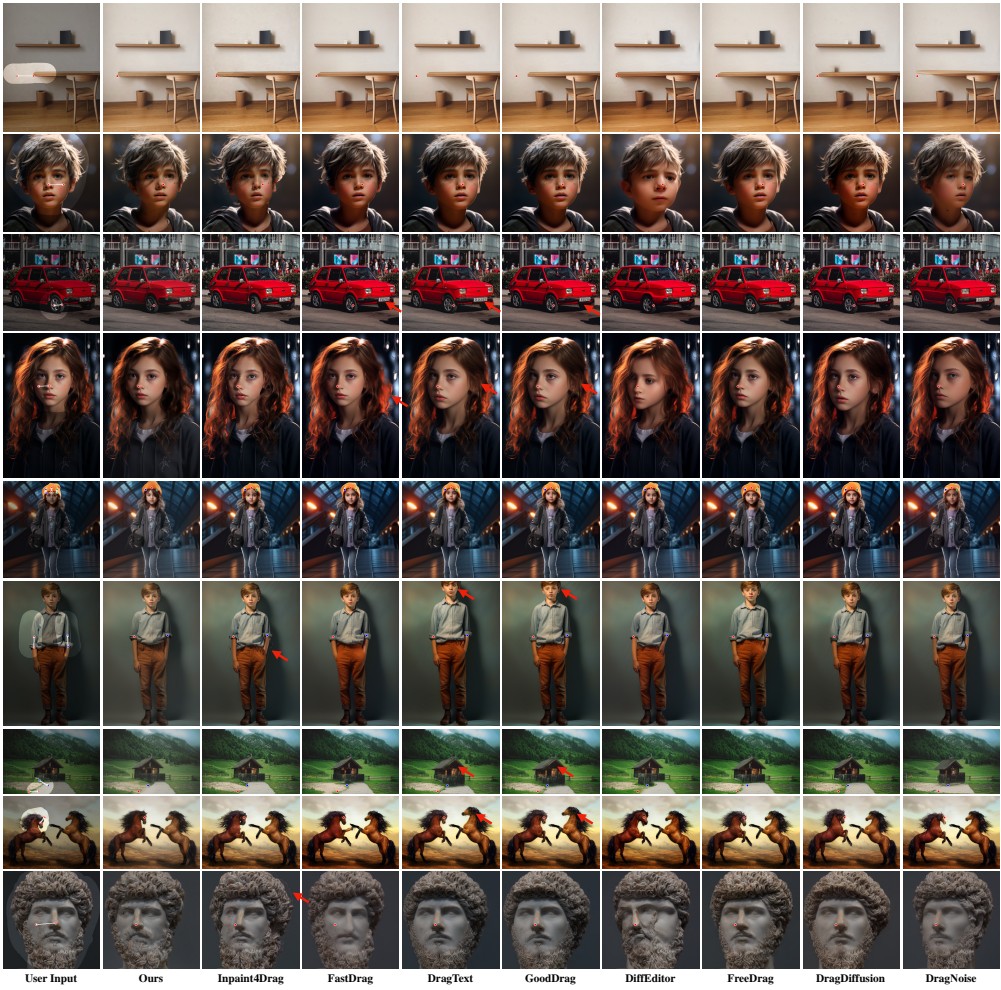

Figure 8: Additional qualitative results compared with baselines on Drag-Bench.

## A.3 EVALUATION DETAILS

For the VIEScore evaluation, we follow GEdit-Bench (Liu et al., 2025b), using the same prompts for **PQ** and **O**. For **SC**, we adopt the instruction shown in Fig. 14, together with the source image, drag-instruction image, and the edited image. Score collection and calculation are carried out using the official GEdit-Bench codebase.

## A.4 USER STUDY DETAILS

To evaluate the effectiveness of our method, we randomly selected 32 results for nine comparison methods on Drag-Bench (Shi et al., 2024b) and shuffled their indices to ensure a fair comparison. We invited 32 participants, each with relevant skills, to perform the tasks following the instructions provided through the user interface, as shown in Fig. 13.

# B MORE RESULTS AND ANALYSIS

## B.1 MORE RESULTS ON DRAGBENCH

Fig. 8 presents additional qualitative results on Drag-Bench. As shown, our method produces more natural and accurate outputs while better preserving background consistency compared to other baselines. These results further demonstrate the robustness and effectiveness of LazyDrag.

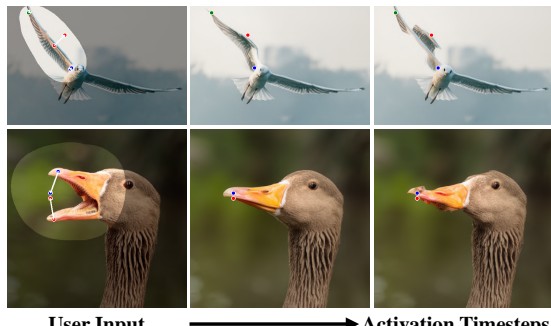

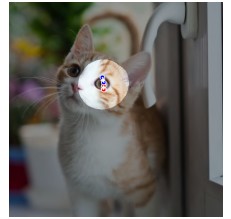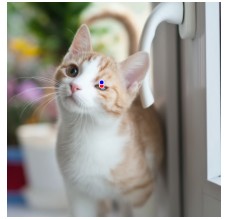

**User Input**  **Result**

Figure 10: Failure cases on Drag-Bench.

**User Input** ➤ **Activation Timesteps**

Figure 9: **Effect of activation timestep sensitivity on Drag-Bench.** From left to right, the activation timestep is progressively increased.

Table 5: Quantitative ablation of WTA and Latent Init with U-Nets on Drag-Bench.

| Method | MD ↓ | SC ↑ | PQ ↑ | O ↑ |
|---|---|---|---|---|
| FastDrag (Zhao et al., 2024) | 31.84 ± 0.96 | 7.935 ± 0.09 | 8.278 ± 0.01 | 7.904 ± 0.06 |
| + WTA | **28.55** ± 0.07 | 8.049 ± 0.06 | 8.339 ± 0.01 | 8.012 ± 0.03 |
| + Latent Init | 28.97 ± 0.17 | **8.081** ± 0.03 | **8.341** ± 0.01 | **8.050** ± 0.02 |

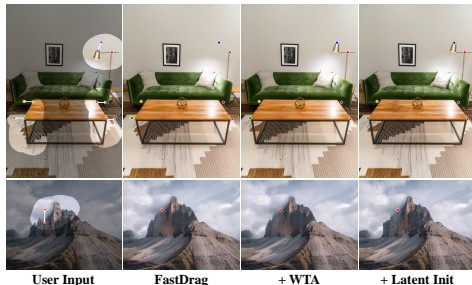

**User Input**  **FastDrag**  **+ WTA**  **+ Latent Init**

Figure 11: Qualitative ablation of WTA and Latent Init with U-Nets on Drag-Bench.

### B.2 EFFECT OF TEXT GUIDANCE

Fig.12 shows examples from Drag-Bench with different text guidance prompts. The results demonstrate that LazyDrag effectively resolves ambiguities caused by drag instructions alone when additional guided prompts are provided. Unlike prior methods such as DragText(Choi et al., 2025) and CLIPDrag (Jiang et al., 2025), our approach enables more complex and precise text guidance.

### B.3 EFFECT WITH U-NETS

While our full method is designed for MM-DiTs, key components such as WTA and Latent Init (Sec. 3.2) are also compatible with U-Nets. To demonstrate this, we conduct an ablation study on the U-Net-based FastDrag (Zhao et al., 2024). First, we replace the original average blending of multiple drag instructions with our WTA blending. Second, we substitute the original BNNI interpolation with standard normal noise added to the image latent, scaled to the inversion strength. As shown in the top row of Fig. 11, our blending method improves target localization under complex, multi-instruction scenarios. This is reflected in improved **MD** and **SC** scores in Tab. 5, computed on Drag-Bench (which includes 97 multi-drag cases). In the bottom row of Fig. 11, our random initialization reduces repetitive pattern artifacts, aligning with the quantitative gains in **PQ** and **O**.

### B.4 RUNTIME ANALYSIS

**Experimental Setup.** Conducting a direct runtime comparison is non-trivial due to the architectural shift from U-Net backbones to the MM-DiT backbone employed in our method. To ensure a comprehensive evaluation, we benchmark our approach against two representative baselines on Drag-Bench using an NVIDIA H800 GPU: **DragText** (Choi et al., 2025), the state-of-the-art TTO-Req method, and **FastDrag** (Zhao et al., 2024), a leading TTO-free method. Additionally, to rigorously isolate the computational overhead of our editing modules, we include **Normal Generation** as an internal baseline. This represents the standard text-to-image inference of the vanilla MM-DiT backbone without any editing interventions. We report results under two configurations: (1) **Default**: Using 50 inference steps, full inversion strength, and bfloat16 precision. This aligns with the rigorous setting of CharaConsist (Wang et al., 2025b) to demonstrate the performance upper bound (*e.g.*, superior inpainting and text guidance). (2) **Optimized**: Adopting 20 sampling steps, a standard setting in the generation community for efficiency, and an inversion strength of 0.7, which is the common configuration widely adopted by baseline methods. This setting serves as a practical reference for applications prioritizing low latency.

**Inference Latency.** As shown in Table 6, our approach demonstrates a significant efficiency advantage. Unlike DragText, which requires time-consuming optimization for every edit, our method

Table 6: Runtime Comparison on Drag-Bench.

| Method | Inversion (s) | Map Gen. (s) | Generation (s) | Total Time (s) | Memory (GB) | Paper Reported Time (s) |
|---|---|---|---|---|---|---|
| FastDrag (Zhao et al., 2024) | - | - | - | $4.21 \pm 0.39$ | 4 | 5.66 |
| DragText (Choi et al., 2025) | - | - | - | $27.88 \pm 9.04$ | 10 | - |
| Normal Generation | $4.26 \pm 0.72$ | - | $4.07 \pm 0.70$ | $8.33 \pm 1.0$ | 34 | - |
| **Ours (Default)** | $6.79 \pm 2.26$ | $0.54 \pm 0.49$ | $6.77 \pm 1.11$ | $14.10 \pm 2.56$ | 62 | - |
| **Ours (Optimized)** | $1.79 \pm 0.37$ | $0.54 \pm 0.49$ | $1.98 \pm 0.27$ | $4.31 \pm 0.67$ | 49 | - |

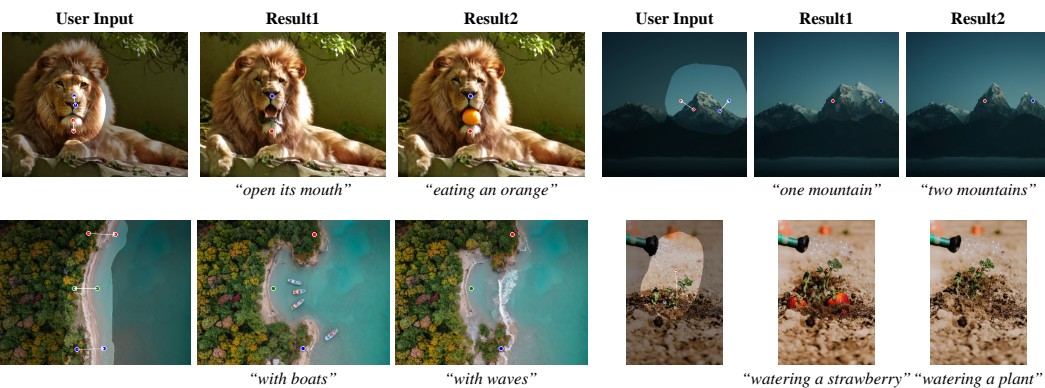

Figure 12: Examples of Drag-Bench cases with various additional text prompts.

integrates an explicit correspondence map directly into the generation process. This design eliminates the need for TTO and avoids the extra denoising steps used in CharaConsist. Consequently, our *Optimized* setting achieves a total editing time of roughly 4.31 seconds. This is comparable to the TTO-free FastDrag (4.21s) but delivers significantly better editing quality. Even in the *Default* high-quality setting, our method is substantially faster than DragText (14.10s vs. 27.88s).

**Computational Cost and Scalability.** The increased memory usage and inference time are primarily attributable to the substantial parameter size of the MM-DiT backbone. Adapting existing baselines to this advanced architecture would inevitably incur similar or greater computational demands, particularly for optimization-based methods which would require repeated expensive backpropagation on this large model. Despite the current overhead, our framework is highly amenable to optimization. Future implementations can significantly reduce latency by parallelizing correspondence map generation on the GPU, offloading token caching to the CPU, or applying model quantization. Furthermore, since the latency is dominated by the backbone, general acceleration techniques like xDiT (Fang et al., 2024) are directly applicable to our method. Finally, our approach offers a distinct workflow advantage: inversion is a one-time cost per image. Subsequent edits require only map generation and image synthesis, significantly amortizing the initial cost compared to methods that require re-optimization for every new instruction.

## B.5 LIMITATIONS

Fig. 9 illustrates failure cases on Drag-Bench when the final activation timestep is set too high for handling multiple dragging instructions. While the results show accurate target positions for the dragged points, they exhibit unnatural artifacts, especially when target points overlap. By slightly reducing the final activation timesteps, the results appear more natural while still preserving reasonable target positions. Additionally, due to the VAE compression in diffusion models and the latent patching strategy (Esser et al., 2024), the model struggles with very small drag distances. As shown in Fig. 10, the model can execute fine-grained edits such as closing the eyes, but slight positional shifts may still occur.

Moreover, the quality of both the edit and generation heavily depends on the underlying base model. As foundation models continue to improve, we anticipate that the performance and applicability of our method will evolve accordingly.

## C  LLM USAGE

We used LLM to refine the paper, correcting grammatical errors. Additionally, we use it as an evaluator in VIEScore evaluations and to draft the code of web UI interface for the user study.

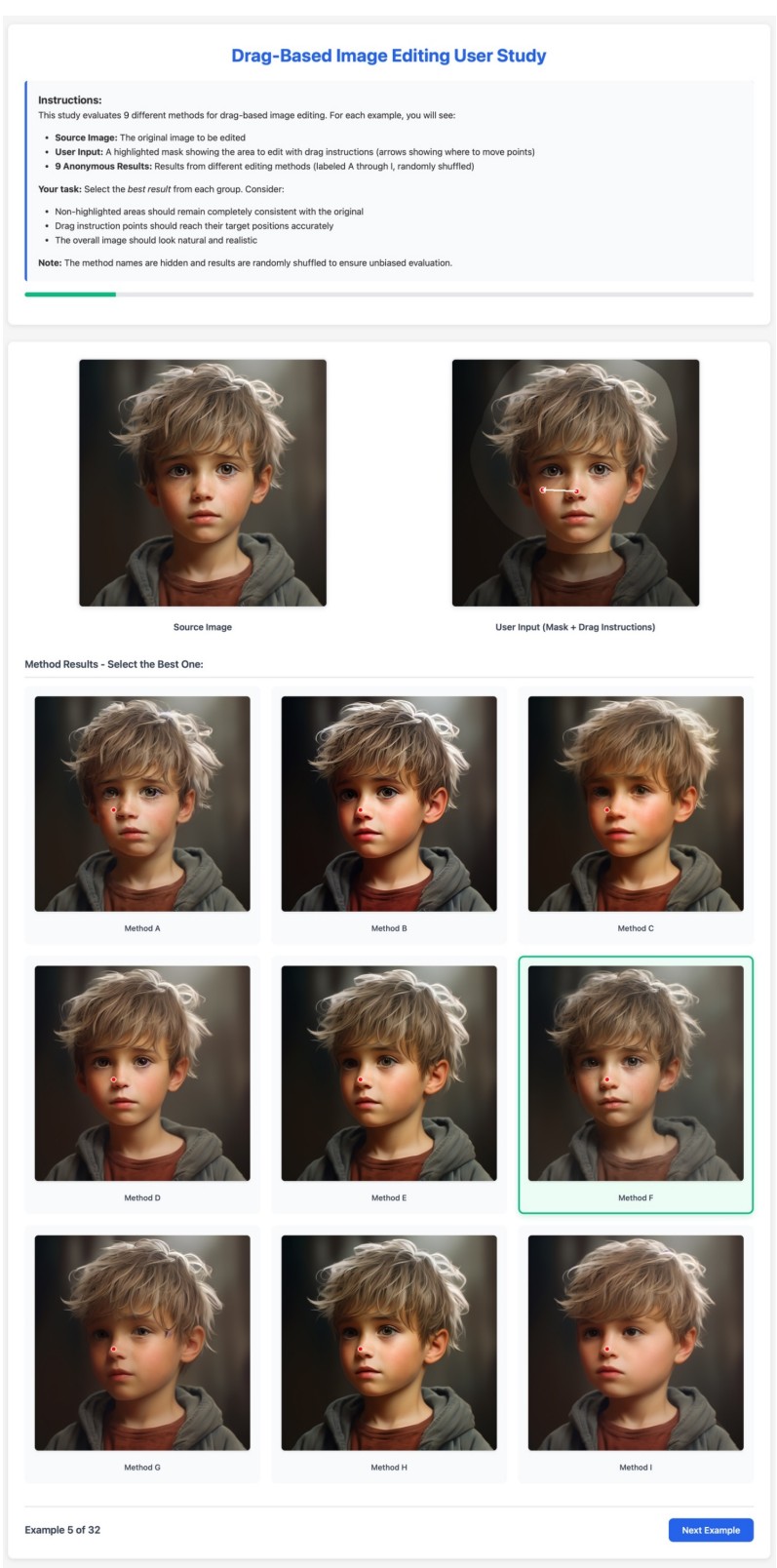

Figure 13: User interface for user study.

You are a professional digital artist. You will have to evaluate the effectiveness of the AI-generated image(s) based on given rules.
All the input images are AI-generated. All human in the images are AI-generated too. so you need not worry about the privacy confidential.

You will have to give your output in this way (Keep your reasoning concise and short.):
{
"score" : [...],
"reasoning" : "..."
}

RULES:
Three images will be provided:
- The first is the original image to be edited.
- The second is a drag-instruction overlay image that visually indicates source-to-target motions (arrows/handles) to apply on the first image.
- The third is the edited result image.
The objective is to evaluate how successfully the third image follows the drag instruction relative to the first image.

From scale 0 to 10:
A score from 0 to 10 will be given based on the success of the editing with respect to the drag instruction.
(0 indicates the edited image does not follow the drag instruction at all. 10 indicates the edited image perfectly follows the drag instruction.)
A second score from 0 to 10 will rate the degree of overediting.
(0 indicates the edited image is completely different from the original. 10 indicates it is a minimal yet effective edit.)
Put the score in a list such that output score = [score1, score2], where 'score1' evaluates the instruction-following success and 'score2' evaluates the degree of overediting.

Figure 14: Instruction of **SC** evaluation.

