# OpenReview forum: "LazyDrag: Enabling Stable Drag-Based Editing on Multi-Modal Diffusion Transformers via Explicit Correspondence"
_ICLR.cc/2026/Conference — ICLR 2026 Poster_

### Official Review · Reviewer_FWPs · 2025-10-28

**Soundness:** 2
**Presentation:** 2
**Contribution:** 2
**Rating:** 6
**Confidence:** 5

**Summary:**

LazyDrag is a training-free drag-based image editing method for Multi-Modal Diffusion Transformers. It replaces unstable attention-based point matching with an explicit correspondence map derived from user drags, which is very similar to Latent Warpage Function of FastDrag, enabling stable full-strength inversion without test-time optimization. This design preserves identity, maintains background consistency, and supports text-guided edits for complex transformations like movement or scaling. Evaluated on DragBench, LazyDrag achieves state-of-the-art accuracy and perceptual quality while remaining efficient and robust, demonstrating that explicit geometric correspondence can unify precise spatial control and semantic guidance in diffusion-based editing.

**Strengths:**

1 The method explicitly associates point matching with the user’s drag instructions following latent warpage function, removing the uncertainty of attention-based implicit matching. This design eliminates many stability and consistency issues seen in previous drag-editing approaches.


2 LazyDrag achieves state-of-the-art results on DragBench across all major metrics, demonstrating both higher drag accuracy and better perceptual quality without requiring test-time optimization.

**Weaknesses:**

1: Computational complexity not included. The paper does not provide quantitative analysis of runtime or memory costs. Since attention modification is applied across multiple layers and steps, its efficiency in large-scale or real-time scenarios is unclear.

2: Limited novelty beyond integration. While the explicit correspondence map is effective, serveral components build upon existing ideas from FastDrag in method section but not discussed in the intro sec. The contribution may be seen as a strong integration rather than a fundamentally new mechanism.

Anyway, I still view it as a meaningful and well-executed improvement. So my suggetion is the authors could explicitly position their contribution as the first to replace implicit point matching with an explicit correspondence map, and visually emphasize this transition and its advantages in the Introduction (e.g., replace the current Figure 1 with a more conceptual figure illustrating this replacement and maybe with a table which shows the adavantages, and move the current example figure to the next page). Secondly, in the method section, a concise side-by-side illustration figure comparing FastDrag’s and LazyDrag’s formulations would also help readers immediately see where the improvements lie. A good example is the left part of Fig1 in https://arxiv.org/pdf/2503.20783.

3: Experiments are mainly conducted on FLUX-based MM-DiT and a single benchmark.

**Questions:**

The paper reports a user study with 20 expert participants evaluating 32 samples each. Could the authors clarify whether this sample size is sufficient to ensure statistical significance and generalization?

---

> ### Author Response · Authors · 2025-11-19
> **Response to Reviewer FWPs (Part 1/N)**
>
> ## Response 1: Comparative results confounded by backbone differences
> Thanks for your thoughtful concerns. It helps us better highlight the core contribution of our method. We clarify that the shift to MM-DiT is not merely for a stronger backbone, but an architectural necessity. Our core contribution (Explicit Correspondence Map) relies on the Rotary Positional Embeddings (RoPE) and attention mechanisms specific to MM-DiTs to resolve the instability of implicit point matching. This mechanism is non-trivial to port to U-Nets, which lack comparable positional embedding designs and text–vision fusion attention strategies.
>
> To isolate the algorithmic gain from the backbone, we point to the ablation study in Table 3 and Figure 6 (the setting with “– D Pres., – Attn Refine”). In this setting, we use the same MM-DiT backbone but remove our Explicit Correspondence Map, reverting to implicit point matching as in prior works as described in L427. The resulting performance degradation (e.g., identity loss, inaccurate dragging) confirms that the improvements come from our proposed editing strategy rather than the backbone itself. Additionally, in Appendix B.3, we ported the transferable components (WTA, Latent Init) to a U-Net baseline (FastDrag), where they also show consistent gains.
> ## Response 2: Limited novelty beyond integration
> Thank you for the encouraging evaluation and for the suggestion on restructuring the introduction. Our introduction is written for readers who may not be familiar with drag-based editing. Many do not realize that implicit point matching is the main cause of instability in prior methods, so we first explain this fundamental issue and then contrast with FastDrag. This allows non-expert readers to understand why introducing an explicit correspondence map is necessary.
>
> We agree that expert readers would benefit from a more direct comparison. However, once the explicit correspondence map is introduced, several components must change accordingly, including latent initialization and attention control. These adjustments make it difficult to summarize the differences in a single figure. For this reason, we adopted the current structure.
> ## Response 3: Generalization
> Thank you for the suggestion. We appreciate the feedback on generalization and will explore additional MM-DiT-based models and benchmarks in future work.
> ## Response 4: User study sample size
> Thank you for the helpful suggestion. We reviewed all baselines and found that only GoodDrag conducted a user study, with 27 participants and 12 samples per participant. Following their setup and your recommendation, we expanded our user study to **32 participants**, while keeping the number of test samples the same (32). The updated results are reported in Table 2. The larger participant pool further supports the reliability and superiority of our method.

---

> > ### Author Response · Authors · 2025-11-27
> > **Thank you for the review!**
> >
> > We sincerely appreciate the time and effort you have dedicated to reviewing our work.
> >
> > We hope you have had a chance to look over our response submitted on November 19. In that response, we provided clarifications regarding results potentially confounded by backbone difference, discussed the novelty of our approach, and updated the sample size of the user study.
> >
> > If there are any remaining concerns or questions, we would be more than happy to provide further clarification.
> >
> > Thank you again for your time and consideration.
> >
> > The Authors

---

### Official Review · Reviewer_xRya · 2025-10-30

**Soundness:** 3
**Presentation:** 2
**Contribution:** 2
**Rating:** 4
**Confidence:** 3

**Summary:**

The paper introduces LazyDrag, a training-free, drag-based image editing method designed for Multi-Modal Diffusion Transformers (MM-DiTs). The core problem it addresses is the instability of prior methods, which rely on implicit point matching via attention, forcing them to use test-time optimization (TTO) or weakened inversion strength. LazyDrag's key contribution is to replace this fragile implicit matching with an explicit correspondence map derived directly from user drag instructions. This deterministic guidance enables stable editing under full-strength inversion without TTO, unlocking high-fidelity inpainting and robust text-guided control. The method is shown to outperform existing baselines on the DragBench benchmark in both accuracy and perceptual quality.

**Strengths:**

1.  **Novel Application and Success on DiT Architectures:** This is arguably the first work to successfully and robustly implement drag-based editing on the emerging MM-DiT architecture. As the field moves from U-Nets to Transformers, this paper provides a timely and valuable contribution, demonstrating a viable path for precise spatial control on these powerful new backbones.

2.  **Principled Solution to a Fundamental Problem:** The authors correctly identify a core bottleneck in previous drag-editing methods—the unreliability of implicit attention-based point matching. Their proposed solution, an explicit correspondence map, is a direct and elegant way to solve this issue. This allows them to eliminate the need for costly test-time optimization (TTO) and, more importantly, enables stable full-strength inversion, which is a significant step forward for the task.

3.  **Impressive Qualitative and Quantitative Results:** The method demonstrates state-of-the-art performance on the DragBench benchmark. The qualitative results are compelling, showcasing complex edits (e.g., opening a mouth and inpainting the interior, generating new objects with text guidance) that prior methods struggle with. The quantitative improvements in mean distance and VIEScore, despite requiring no TTO, strongly support the effectiveness of the proposed approach.

**Weaknesses:**

1.  **Confounded Comparison of Foundation Models:** The primary quantitative results claim state-of-the-art performance by comparing the proposed method, built on a powerful MM-DiT backbone (FLUX.1), against baselines developed for older U-Net architectures (e.g., Stable Diffusion v1.5). This comparison is confounded, as it's unclear how much of the performance improvement is due to the novel LazyDrag algorithm versus the superior generative capabilities of the underlying foundation model. A more rigorous evaluation would involve comparing the methods on the same backbone to isolate the algorithmic contribution.

2.  **Insufficient Differentiation from Prior Motion-Guided Methods:** The core idea of using an explicit motion field to guide a training-free editing process has precedents in recent literature (e.g., InstantDrag, LightningDrag). While the paper's specific implementation—transforming this field into a correspondence map to directly control attention is novel, the manuscript would be stronger if it more clearly articulated this distinction. A direct, mechanistic comparison of how LazyDrag *uses* the motion information versus how prior works use it (e.g., for latent warping or loss-based supervision) is needed to better situate the paper's unique contribution.

3.  **Sensitivity to Key Hyperparameters:** The method's effectiveness appears highly sensitive to the `activation timestep` hyperparameter, as revealed in the paper's own ablation study (Sec. 4.6). The results show a stark trade-off: lower values fail to move points accurately, while higher values introduce significant artifacts, especially with complex instructions. This implies that while the method avoids test-time optimization, it may require careful, per-edit manual tuning to achieve satisfactory results, which somewhat tempers the "lazy" and automated nature of the proposed workflow.

**Questions:**

Could you elaborate on the failure modes of the "winner-takes-all" strategy for displacement field calculation? For instance, how does it behave with a high density of conflicting drag points in a small region, and what are the computational implications?

---

> ### Author Response · Authors · 2025-11-19
> **Response to Reviewer xRya (Part 1/N)**
>
> ## Response 1: Comparative results confounded by backbone differences
> Thanks for your thoughtful concerns. It helps us better highlight the core contribution of our method. We clarify that the shift to MM-DiT is not merely for a stronger backbone, but an architectural necessity. Our core contribution (Explicit Correspondence Map) relies on the Rotary Positional Embeddings (RoPE) and attention mechanisms specific to MM-DiTs to resolve the instability of implicit point matching. This mechanism is non-trivial to port to U-Nets, which lack comparable positional embedding designs and text–vision fusion attention strategies.
>
> To isolate the algorithmic gain from the backbone, we point to the ablation study in Table 3 and Figure 6 (the setting with “– D Pres., – Attn Refine”). In this setting, we use the same MM-DiT backbone but remove our Explicit Correspondence Map, reverting to implicit point matching as in prior works as described in L427. The resulting performance degradation (e.g., identity loss, inaccurate dragging) confirms that the improvements come from our proposed editing strategy rather than the backbone itself. Additionally, in Appendix B.3, we ported the transferable components (WTA, Latent Init) to a U-Net baseline (FastDrag), where they also show consistent gains.
> ## Response 2: Differentiation from motion-guided methods
> Thanks for pointing this out. InstantDrag and LightningDrag are strong works, and we have added them to the related work. To clarify, InstantDrag and LightningDrag are **not training-free approaches**. Both methods rely on **additional model training**, whereas our method is **fully training-free** and operates directly on pre-trained MM-DiT backbones.
>
> Latent warping is only an initialization and is fundamentally insufficient for stable editing. Even with the best possible warping, it suffers from the inherent limitation described in L196: after initialization, the handle-specific cues vanish and all moved regions are treated uniformly, creating a trade-off between positional accuracy and naturalness. Our results also show this limitation. Table 5 and Figure 11 demonstrates that even FastDrag benefits markedly when combined with our stronger latent initialization, confirming that latent warping alone cannot solve the core instability.
>
> Loss-based supervision is also impractical for interaction. As shown in our new Appendix B.5 timing analysis, these methods run slower on a U-Net than our training-free method runs on a much larger MM-DiT, and they must re-optimize for every user edit, leading to poor user experience.
>
> In contrast, our Explicit Correspondence Map directly resolves the fundamental implicit point matching problem. This enables capabilities that prior works cannot support, including strong text guidance, robust inpainting, and natural high-quality edits.
> ## Response 3: Sensitivity to Hyperparameters
> Thanks for your careful observation. All benchmark cases use the same ``activation_timestep`` of 40, as stated in L468. We do not perform any per-edit manual tuning. Under this unified setting, our method already produces strong results. Other baselines cannot achieve comparable quality even with TTO or per-instruction optimization.
>
> The artifacts in this example are not caused by complex instructions. They come from ambiguous benchmark prompts. We follow the original benchmark instructions without any manual adjustment, unlike methods such as Inpaint4Drag. Prior work (e.g., DragText) and our own results in Fig. 1 and Fig. 12 show that clearer text descriptions provide significant benefits for complex edits. Most benchmark prompts only describe the object rather than the intended transformation. With richer descriptions, our method can flexibly produce effects such as widening a mountain or splitting it into two narrow peaks in Figure 12.
>
> This trade-off between instruction ambiguity and editing precision is inherent to drag-based editing. Our method still performs far better than all baselines. Even under the extreme ``activation_timestep`` of 50, our overall score of 8.087 surpasses every baseline.

---

> ### Author Response · Authors · 2025-11-19
> **Response to Reviewer xRya (Part 2/N)**
>
> ## Response 4: Failure modes of WTA
> Thanks for your thoughtful question. Our method does not achieve pixel-level precision in very small spatial regions, as shown in Figure 10 of Appendix B.5. Due to VAE compression and the MM-DiT patchifying operation, the dragging accuracy has an inherent tolerance of about 16 pixels.
>
> This limitation mainly affects extremely small-range edits. In normal cases, high-density but reasonable drags do not cause noticeable issues. For example, in the top row of Figure 1, the upper and lower lips are dragged in opposite directions from nearly the same starting points, yet the results remain clean. Points outside the dense region obtain accurate and unambiguous correspondences through WTA.
>
> Inside high-density regions, the behavior depends on the drag pattern. When points diverge outward, the model naturally creates an inpainting region and fills it with coherent content, such as the tongue and teeth of the dog in Figure 1. When multiple points incorrectly collapse toward the same target, as in the bottom row of Fig. 9, using excessively high ``activation_timestep`` can cause compression artifacts. However, with a reasonable activation value (we use the same value of 40 throughout the paper), or with properly placed starting points at object boundaries, the edits remain stable and natural, as demonstrated in Fig. 9.

---

> > ### Author Response · Authors · 2025-11-27
> > **Thank you for the review!**
> >
> > We sincerely appreciate the time and effort you have dedicated to reviewing our work.
> >
> > We hope you have had a chance to look over our response submitted on November 19. In that response, we provided clarifications regarding results potentially confounded by backbone difference, discussed the differentiation from motion-guided methods, analyzed the sensitivity to hyperparameters, and examined the failure modes of WTA.
> >
> > If there are any remaining concerns or questions, we would be more than happy to provide further clarification.
> >
> > Thank you again for your time and consideration.
> >
> > The Authors

---

### Official Review · Reviewer_ruF6 · 2025-11-01

**Soundness:** 2
**Presentation:** 3
**Contribution:** 2
**Rating:** 6
**Confidence:** 3

**Summary:**

The paper proposes LazyDrag, a training-free drag-based image editing method for MM-DiTs that replaces point matching with an explicit correspondence map derived from user drags. This design enables stable full-strength inversion without test-time optimization. LazyDrag supports multi-round operations (move/scale) and resolves ambiguous drags via prompt guidance, delivering higher drag accuracy and visual quality than prior baselines on DragBench and in user studies. From the paper, this is: (1) the first MM-DiT drag editor under full-strength inversion, (2) explicit correspondence-driven attention controls that remove TTO, and (3) SOTA results with text-guided, natural inpainting.

**Strengths:**

* From the experiments, this method achieves the state-of-the-art performance in DragBench (Table 1), as well as achieving solid empirical results with user preference. In addition, this paper also integrates editing, like scaling and movement. Although these operations are relatively common in latent editing, it's good to show such ability in one pipeline.
* Compared with optimization-based methods like Dragdiffusion and DragGAN, the proposed method provides faster inference speed.
* The overall pipeline is clearly illustrated and easy to understand.

**Weaknesses:**

* The core idea of explicit drag-to-correspondence plus attention control feels incremental relative to prior drag/edit pipelines. Specifically, the TTO-free idea has been used in many previous works (like SDE-Edit, RegionDrag), although the way of replacement may be different. The full-strength editing is also natural in the inversion-based editing. From this perspective, the technical novelty of this paper is not strong. The abstract and relative part needs to claim the core contribution more carefully and clearly.
* Although the paper shows significant improvements in Table 1, the metrics' increase seems much more marginal in Table 5, meaning that the major improvement factor comes from changing to DiT. Since it makes such a big difference, it is helpful to remind of this detail in Table 1, which shows both cases (UNet and DiT).
* A detailed memory and time cost of the proposed method is missing, especially when the paper emphasizes the efficiency of TTO-free design.

**Questions:**

* The quantitative evaluation seems impressive, but why does the qualitative evaluation seem worse than the baselines? E.g., In Figure 8 last row, the man's face seems distorted compared with DragNoise.

---

> ### Author Response · Authors · 2025-11-19
> **Response to Reviewer ruF6 (Part 1/N)**
>
> ## Response 1: Novelty of TTO-free design
> Thank you for the helpful feedback. Full inversion and TTO-free editing indeed offer strong advantages such as better inpainting, stronger text guidance, and more natural edits. However, prior works cannot reliably support these features because **implicit point matching is fundamentally unstable**, which forces them to either lower the inversion strength or rely on TTO to suppress artifacts.
>
> Our contribution is to identify and resolve this core limitation. The Explicit Correspondence Map replaces unreliable implicit point matching, allowing stable full-strength inversion while maintaining edit quality. This makes TTO-free pipelines feasible without sacrificing identity preservation or edit accuracy.
>
> As discussed in the paper, baselines are forced into low-strength inversion or TTO for two reasons:
> 1. Implicit attention-based matching allocates weights to nearby pixels rather than semantically corresponding ones, leading to drift and identity loss (L36).
> 2. Latent warping alone cannot balance positional accuracy and natural inpainting, as the handle-specific cues are lost after initialization (L196).
>
> In short, existing methods are not overlooking full inversion or TTO-free design. They simply cannot support them due to the inherent limitations of implicit matching. By solving this problem, our method makes these desirable features finally achievable.
> ## Response 2: Marginal gains in Table 5 vs Table 1
> Thank you for the suggestion. We have further emphasized in the experimental section that the baselines use U-Net while our method is built on MM-DiT. Table 5 evaluates only the components (WTA, Latent Init) transferable to U-Net. It does not include our core contribution (Explicit Correspondence Map), which is MM-DiT specific. It relies on the Rotary Positional Embeddings (RoPE) and attention mechanisms specific to MM-DiTs to resolve the instability of implicit point matching. Table 1 reflects the performance of our full method and shows significant improvements.
>
> To isolate the algorithmic gain from the backbone, we point to the ablation study in Table 3 and Figure 6 (the setting with “– D Pres., – Attn Refine”). In this setting, we use the same MM-DiT backbone but remove our Explicit Correspondence Map, reverting to implicit point matching as in prior works as described in L427. The resulting performance degradation (e.g., identity loss, inaccurate dragging) confirms that the improvements come from our proposed editing strategy rather than the backbone itself.
> ## Response 3: Runtime & efficiency
> We address the efficiency concerns with a detailed analysis provided in Appendix B.4. Since MM-DiTs are inherently more computationally demanding than U-Nets, we evaluate efficiency not only against state-of-the-art TTO-Req methods (e.g., DragText) and the leading TTO-free baseline (FastDrag), but also relative to a Normal Generation baseline (standard text-to-image inference) on an NVIDIA H800 GPU. We report results under two configurations: a Default setting to demonstrate the performance upper bound, and an Optimized setting for practical latency. Notably, the latter adopts 20 sampling steps (a community standard) and 0.7 inversion strength (widely used by baselines) to ensure a fair and realistic efficiency comparison.
> 1. **Interactive Latency**. As shown in the table below, our optimized configuration achieves a total latency of 4.31s, comparable to FastDrag (4.21s). Even our default setting (14.10s) is substantially faster than DragText (27.88s), demonstrating that our method remains viable for interactive editing despite using a larger backbone.
> 2. **Backbone Cost vs. Method Overhead**. The increased memory footprint primarily arises from the 12B-parameter MM-DiT backbone rather than from our editing module. The additional latency introduced by our method beyond the backbone’s forward pass is modest and well-contained.
> 3. **Amortized Cost**. Importantly, our method performs inversion only once per image. Subsequent edits reuse the inverted latents and require only correspondence map generation and image synthesis, whereas TTO-Req methods must perform optimization for every edit.
> | Method | Inversion (s) | Map Gen (s) | Generation (s) | Total Time (s) | Memory (GB) | Paper Reported Time (s)
> | :--- | :---: | :---: | :---: | :---: | :---: | :---: |
> | FastDrag (U-Net) | - | - | - | 4.21 $\pm$  0.39| 4 | 5.66 |
> | DragText (U-Net) | - | - | - | 27.88 $\pm$ 9.04 | 10 | - |
> | Normal Generation (MM-DiT) | 4.26 $\pm$  0.72 | - | 4.07 $\pm$  0.70  | 8.33 $\pm$  1.0  | 34 | - |
> | Ours (Default) | 6.79 $\pm$   2.26  | 0.54 $\pm$  0.49  | 6.77 $\pm$  1.11 | 14.10 $\pm$   2.56  | 62 | - |
> | Ours (Optimized) | 1.79 $\pm$  0.37 | 0.54 $\pm$   0.49  | 1.98 $\pm$   0.27 | 4.31 $\pm$   0.67 | 49 | -|

---

> ### Author Response · Authors · 2025-11-19
> **Response to Reviewer ruF6 (Part 2/N)**
>
> ## Response 4: Qualitative evaluation
> Thank you for the careful observation. DragNoise has always been a special case among prior methods: it tends to produce more natural-looking results, but its identity shift is usually the most severe and it is also more prone to artifacts. For example, the source image has a smooth reflective surface, while its output turns into a matte texture.
>
> We followed the original benchmark prompts and did not tune the text prompts for individual cases. In real applications, users can add guidance such as “turn head” to help achieve more accurate results.
>
> In addition, as shown in Table 4 and Figure 7, our method includes an activation timestep that adjusts the trade-off between drag accuracy and naturalness. Throughout the paper, we used a single fixed balance value, while practical usage would typically adopt a more natural setting.

---

> > ### Author Response · Authors · 2025-11-27
> > **Thank you for the review!**
> >
> > We sincerely appreciate the time and effort you have dedicated to reviewing our work.
> >
> > We hope you have had a chance to look over our response submitted on November 19. In that response, we provided clarifications regarding results potentially confounded by backbone difference, discussed novelty of our TTO-free design, and included analyses of runtime and efficiency.
> >
> > If there are any remaining concerns or questions, we would be more than happy to provide further clarification.
> >
> > Thank you again for your time and consideration.
> >
> > The Authors

---

### Official Review · Reviewer_kx3n · 2025-11-03

**Soundness:** 3
**Presentation:** 2
**Contribution:** 3
**Rating:** 6
**Confidence:** 4

**Summary:**

The paper introduces LazyDrag, a drag-based image editing method for DiT-family models (MM-DiTs). It performs full-strength inversion and avoids test-time optimization (TTO). The core idea is to learn an explicit correspondence map guided by text, combined with background and identity preservation to deliver high-quality, controllable drags. The paper provides broad comparisons with prior drag-editing approaches and reports strong qualitative and quantitative results.

**Strengths:**

1.	Modern backbone: Extends drag editing from U-Net diffusion (e.g., Stable Diffusion) to contemporary DiT architectures, which is timely and relevant.
2.	No test time optimization: The method avoids test-time optimization, which can reduce engineering complexity and potentially runtime.
3.	Instruction clarity: The correspondence map makes drag constraints explicit and easier to interpret.
4.	Compelling results: Both qualitative figures and reported metrics indicate performance gains over prior work.

**Weaknesses:**

1.	Runtime and efficiency claims need evidence
No TTO does not automatically imply faster end-to-end editing if additional modules (inversion, correspondence estimation, preservation) are heavy.
Could the authors report end-to-end wall-clock time, GPU memory for LazyDrag vs. other methods? Include all components and required steps related to that method, both TTO free and non-free methods.

2.	The paper lists full-strength inversion as a main contribution. Why is full-strength inversion necessary for drag editing? In Fig. 2, full-strength inversion appears to improve LazyDrag’s results, but is this inherently required by the drag-editing setup? Also, higher inversion strength likely increases computation time.

3.	Please add experiments demonstrating that the performance gains come from the LazyDrag design rather than from the MM-DiT backbone. Prior methods use U-Net models (e.g., Stable Diffusion), while this paper uses FLUX.1 Krea-dev. How do you show that improvements are not dominated by the model shift?

4.	The experiments report Semantic Consistency and Perceptual Quality. Could you add results to justify that these metrics are suitable for drag editing, for example, by also reporting other metrics used in related work and comparing against them?

5.	The stated contributions and the method description feel slightly misaligned. The methods section is divided into two parts, with the second devoted to preservation, but “preservation” is not clearly reflected among the main contributions. Meanwhile, the introduction emphasizes no test-time optimization and full-strength inversion. Clarifying and aligning the concepts and contributions across the introduction and methods would improve the paper.

**Questions:**

1.	This paper treats background and identity differently. How are background and objects detected and distinguished in practice? Are there any additional steps, or do you rely on other methods?
2.	The shift to DiT-based image models is natural, since newer models are generally more powerful than earlier ones. This could bring future benefits to the drag editing community.

---

> ### Author Response · Authors · 2025-11-19
> **Response to Reviewer kx3n (Part 1/N)**
>
> ## Response 1: Runtime & efficiency
> We address the efficiency concerns with a detailed analysis provided in Appendix B.4. Since MM-DiTs are inherently more computationally demanding than U-Nets, we evaluate efficiency not only against state-of-the-art TTO-Req methods (e.g., DragText) and the leading TTO-free baseline (FastDrag), but also relative to a Normal Generation baseline (standard text-to-image inference) on an NVIDIA H800 GPU. We report results under two configurations: a Default setting to demonstrate the performance upper bound, and an Optimized setting for practical latency. Notably, the latter adopts 20 sampling steps (a community standard) and 0.7 inversion strength (widely used by baselines) to ensure a fair and realistic efficiency comparison.
> 1. **Interactive Latency**. As shown in the table below, our optimized configuration achieves a total latency of 4.31s, comparable to FastDrag (4.21s). Even our default setting (14.10s) is substantially faster than DragText (27.88s), demonstrating that our method remains viable for interactive editing despite using a larger backbone.
> 2. **Backbone Cost vs. Method Overhead**. The increased memory footprint primarily arises from the 12B-parameter MM-DiT backbone rather than from our editing module. The additional latency introduced by our method beyond the backbone’s forward pass is modest and well-contained.
> 3. **Amortized Cost**. Importantly, our method performs inversion only once per image. Subsequent edits reuse the inverted latents and require only correspondence map generation and image synthesis, whereas TTO-Req methods must perform optimization for every edit.
> | Method | Inversion (s) | Map Gen (s) | Generation (s) | Total Time (s) | Memory (GB) | Paper Reported Time (s)
> | :--- | :---: | :---: | :---: | :---: | :---: | :---: |
> | FastDrag (U-Net) | - | - | - | 4.21 $\pm$  0.39| 4 | 5.66 |
> | DragText (U-Net) | - | - | - | 27.88 $\pm$ 9.04 | 10 | - |
> | Normal Generation (MM-DiT) | 4.26 $\pm$  0.72 | - | 4.07 $\pm$  0.70  | 8.33 $\pm$  1.0  | 34 | - |
> | Ours (Default) | 6.79 $\pm$   2.26  | 0.54 $\pm$  0.49  | 6.77 $\pm$  1.11 | 14.10 $\pm$   2.56  | 62 | - |
> | Ours (Optimized) | 1.79 $\pm$  0.37 | 0.54 $\pm$   0.49  | 1.98 $\pm$   0.27 | 4.31 $\pm$   0.67 | 49 | -|
> ## Response 2: Necessity of full-strength inversion
> Thanks for your good question. Full-strength inversion is a capability, not a hard constraint.
> - **Why it matters**: Prior methods **must** use lower strength or TTO to avoid artifacts caused by unreliable implicit matching. This compromise limits their ability to perform inpainting (filling disoccluded regions) and follow text guidance.
> - **Our Advantage**: Because our Explicit Correspondence Map resolves the matching problem, we can safely use full strength. This enables superior inpainting and stronger text controllability, as shown in Fig. 2. Users who prioritize speed or whose edit task is relatively simple may still choose a lower strength if desired.
> ## Response 3: Comparative results confounded by backbone differences
> Thank you for the insightful comment. It helps us better highlight the core contribution of our method. We clarify that the shift to MM-DiT is not merely for a stronger backbone, but an architectural necessity. Our core contribution (Explicit Correspondence Map) relies on the Rotary Positional Embeddings (RoPE) and attention mechanisms specific to MM-DiTs to resolve the instability of implicit point matching. This mechanism is non-trivial to port to U-Nets, which lack comparable positional embedding designs and text–vision fusion attention strategies.
>
> To isolate the algorithmic gain from the backbone, we point to the ablation study in Table 3 and Figure 6 (the setting with “– D Pres., – Attn Refine”). In this setting, we use the same MM-DiT backbone but remove our Explicit Correspondence Map, reverting to implicit point matching as in prior works as described in L427. The resulting performance degradation (e.g., identity loss, inaccurate dragging) confirms that the improvements come from our proposed editing strategy rather than the backbone itself. Additionally, in Appendix B.3, we ported the transferable components (WTA, Latent Init) to a U-Net baseline (FastDrag), where they also show consistent gains.

---

> ### Author Response · Authors · 2025-11-19
> **Response to Reviewer kx3n (Part 2/N)**
>
> ## Response 4: Metrics (SC and PQ)
> Thank you for your valuable feedback. SC and PQ are widely adopted as core evaluation metrics in recent image editing works such as Qwen-Image, Bagel, OmniGen2, Step1X-Edit, and EMU3.5. PQ measures only perceptual image quality and is independent of the editing task, and we follow exactly the same evaluation setup. The original SC metric is designed for general image editing evaluation, and we only add drag-specific descriptions while keeping all other components unchanged, as shown in Fig. 14. Moreover, GoodDrag also emphasizes that LLM-based judgments are more reliable, and LLM-based evaluation has become standard practice in recent image editing approaches. To further validate our effectiveness, we additionally include a user study in Tab. 2.
>
> Traditional drag-editing metrics primarily include IF and MD. We report MD. The IF metric is excluded due to its known unreliability. Both DragText and RegionDrag explicitly note that IF has limited evaluative value (see line 355). Specifically, models that perform precise dragging often induce larger local changes and therefore obtain worse IF scores, while a method that makes no edits at all can theoretically achieve the best IF score (1). This makes IF misleading as an evaluation metric.
> ## Response 5: Alignment of contributions
> Thank you for the suggestion. Our method section is organized around first generating the explicit correspondence map and then using it for preservation. Based on your suggestion, we have already **refined the contribution in the revision**.
>
> Regarding the introduction style, we designed it for readers who may not know that unstable implicit point matching is the root cause behind the reliance on TTO and low-strength inversion in prior work. We therefore first explain how low inversion strength suppresses key generative abilities, then show that avoiding both TTO and low strength requires solving implicit point matching. This leads to the core observation that only MM-DiT, with its RoPE-based attention, allows explicit correspondence maps to be effective, unlike U-Nets. The logical chain is: explicit correspondence map enables no TTO and full-strength inversion, no TTO and full-strength inversion restores high-quality inpainting and natural edits. This narrative emphasizes the progression of ideas rather than listing technical differences directly, which may feel less direct for expert readers like you.
> ## Response 6: Background and object distinction
> In our current setting, the background and identity regions are defined by the user-provided mask that already comes with the DragBench benchmark. Operationally, we follow FastDrag and use the circumcircle of the bounding box of the mask to separate foreground from background (L788 in the Appendix). Since DragBench masks are roughly hand-drawn, the model does not rely on precise mask boundaries. In real applications, users can simply click to invoke SAM or other segmentation models to obtain accurate masks.

---

> > ### Comment · Reviewer_kx3n · 2025-11-24
> >
> > Thank you for your response. It helped clarify some concerns. I will maintain my recommendation for acceptance.

---

> > > ### Author Response · Authors · 2025-11-25
> > > **Appreciation for the constructive comments**
> > >
> > > We sincerely appreciate your helpful suggestions! Thanks for your time and effort!

---

### Official Review · Reviewer_fc7b · 2025-11-04

**Soundness:** 3
**Presentation:** 3
**Contribution:** 3
**Rating:** 6
**Confidence:** 4

**Summary:**

This paper presents LazyDrag, a training-free drag-based image editing method tailored to multi-modal diffusion transformers. The core idea is to replace implicit attention-based point matching with an explicit correspondence map computed from user drag instructions. This map drives both latent initialization in uncovered regions and attention control via token replacement and gated blending, which allows edits under strong inversion while preserving image identity. Experiments on a standard drag-editing benchmark and several recent baselines show improved spatial accuracy, semantic consistency, and perceptual quality without test-time optimization. A user study further indicates that human raters prefer LazyDrag in most cases, suggesting practical relevance for interactive editing.

**Strengths:**

- Clear diagnosis of why attention-similarity–based matching fails under strong inversion, which directly motivates the shift to explicit correspondence maps and gives the method a coherent conceptual foundation.

- Elegant use of a single correspondence map to control both latent initialization and attention, enabling principled handling of background, inpainting, and transition regions and reducing artifacts from naïve displacement blending.

- Strong emprical results on a standard drag-editing benchmark, with consistent gains in spatial accuracy, semantic consistency, and perceptual quality over several recent drag-based baselines, and without test-time optimization.

- Qualitative and user-study evidence that the method handles complex and text-guided edits (e.g. ambiguous drags, newly revealed content) in a way that is preferred by human raters.

**Weaknesses:**

- Comparative results are confounded by differences in backbone and inversion quality, because LazyDrag runs on a strong MM-DiT while several baselines still use UNet models. Implementing LazyDrag on a shared U-Net backbone or porting at least one baseline to the same MM-DiT would clarify how much of the gain comes from the editing strategy itself.

- Efficiency claims remain qualitative, even though interactive editing hinges on latency and resource usage; reporting wall-clock runtime and memory consumption against speed-oriented methods would make the practical advantages more credible.

- Generalization beyond a single MM-DiT backbone and benchmark is not demonstrated.

- The text-guided resolution of ambiguous drags is mainly shown through qualitative examples. Maybe a small controlled benchmark or focused user study on this aspect would better quantify its advantage over purely drag-based methods.

**Questions:**

- Can you provide numbers for runtime and memory usage per edit (including inversion) compared to speed-oriented methods such as FastDrag-style or inpainting-based approaches on same hardware?

- Are there architectural assumptions that make LazyDrag significantly easier to implement on MM-DiTs than on U-Net backbones, or is it a design choice?

- How robust is the method to some poorly placed drag points, for instance when both handle and target lie slightly off the intended object or cross semantic boundaries?

---

> ### Author Response · Authors · 2025-11-19
> **Response to Reviewer fc7b (Part 1/N)**
>
> ## Response 1: Comparative results confounded by backbone differences
> Thanks for your thoughtful concerns. It helps us better highlight the core contribution of our method. We clarify that the shift to MM-DiT is not merely for a stronger backbone, but an architectural necessity. Our core contribution (Explicit Correspondence Map) relies on the Rotary Positional Embeddings (RoPE) and attention mechanisms specific to MM-DiTs to resolve the instability of implicit point matching. This mechanism is non-trivial to port to U-Nets, which lack comparable positional embedding designs and text–vision fusion attention strategies.
>
> To isolate the algorithmic gain from the backbone, we point to the ablation study in Table 3 and Figure 6 (the setting with “– D Pres., – Attn Refine”). In this setting, we use the same MM-DiT backbone but remove our Explicit Correspondence Map, reverting to implicit point matching as in prior works as described in L427. The resulting performance degradation (e.g., identity loss, inaccurate dragging) confirms that the improvements come from our proposed editing strategy rather than the backbone itself. Additionally, in Appendix B.3, we ported the transferable components (WTA, Latent Init) to a U-Net baseline (FastDrag), where they also show consistent gains.
> ## Response 2: Runtime & efficiency
> We address the efficiency concerns with a detailed analysis provided in Appendix B.4. Since MM-DiTs are inherently more computationally demanding than U-Nets, we evaluate efficiency not only against state-of-the-art TTO-Req methods (e.g., DragText) and the leading TTO-free baseline (FastDrag), but also relative to a Normal Generation baseline (standard text-to-image inference) on an NVIDIA H800 GPU. We report results under two configurations: a Default setting to demonstrate the performance upper bound, and an Optimized setting for practical latency. Notably, the latter adopts 20 sampling steps (a community standard) and 0.7 inversion strength (widely used by baselines) to ensure a fair and realistic efficiency comparison.
> 1. **Interactive Latency**. As shown in the table below, our optimized configuration achieves a total latency of 4.31s, comparable to FastDrag (4.21s). Even our default setting (14.10s) is substantially faster than DragText (27.88s), demonstrating that our method remains viable for interactive editing despite using a larger backbone.
> 2. **Backbone Cost vs. Method Overhead**. The increased memory footprint primarily arises from the 12B-parameter MM-DiT backbone rather than from our editing module. The additional latency introduced by our method beyond the backbone’s forward pass is modest and well-contained.
> 3. **Amortized Cost**. Importantly, our method performs inversion only once per image. Subsequent edits reuse the inverted latents and require only correspondence map generation and image synthesis, whereas TTO-Req methods must perform optimization for every edit.
>
>
> | Method | Inversion (s) | Map Gen (s) | Generation (s) | Total Time (s) | Memory (GB) | Paper Reported Time (s)
> | :--- | :---: | :---: | :---: | :---: | :---: | :---: |
> | FastDrag (U-Net) | - | - | - | 4.21 $\pm$  0.39| 4 | 5.66 |
> | DragText (U-Net) | - | - | - | 27.88 $\pm$ 9.04 | 10 | - |
> | Normal Generation (MM-DiT) | 4.26 $\pm$  0.72 | - | 4.07 $\pm$  0.70  | 8.33 $\pm$  1.0  | 34 | - |
> | Ours (Default) | 6.79 $\pm$   2.26  | 0.54 $\pm$  0.49  | 6.77 $\pm$  1.11 | 14.10 $\pm$   2.56  | 62 | - |
> | Ours (Optimized) | 1.79 $\pm$  0.37 | 0.54 $\pm$   0.49  | 1.98 $\pm$   0.27 | 4.31 $\pm$   0.67 | 49 | -|
>
> ## Response 3: Generalization
> Thank you for the suggestion. We appreciate the feedback on generalization and will explore additional MM-DiT-based models and benchmarks in future work. Regarding the ambiguous-drag benchmark, we agree that a dedicated evaluation would be valuable and will consider developing it in future iterations of the project.
> ## Response 4: Architectural assumptions
> Thank you for your insightful question. This is an architectural choice driven by necessity. The Explicit Correspondence Map relies on RoPE to modulate attention tokens effectively, a mechanism that U-Nets do not provide. However, as shown in Table 3 and Figure 6, MM-DiTs using their original implicit point matching exhibit clear limitations. Moreover, Appendix B.3 demonstrates that certain sub-components, such as WTA and Latent Init, are transferable to U-Nets, yielding consistent improvements.
>
> ## Response 5: Robustness to poorly placed points.
> As discussed in Fig 9, robustness is controlled by the ``activation_step``. We use a balanced default of 40 steps across all experiments, which provides high precision while remaining tolerant to slight point misplacements (e.g., a handle point not exactly on the boundary). If a user desires extremely precise editing, they may increase the ``activation_step``. However, doing so requires correspondingly higher accuracy in specifying the handle and target points.

---

> > ### Author Response · Authors · 2025-11-27
> > **Thank you for the review!**
> >
> > We sincerely appreciate the time and effort you have dedicated to reviewing our work.
> >
> > We hope you have had a chance to look over our response submitted on November 19. In that response, we provided clarifications regarding results potentially confounded by backbone differences, analyses of runtime and efficiency, discussions of architectural assumptions and robustness.
> >
> > If there are any remaining concerns or questions, we would be more than happy to provide further clarification.
> >
> > Thank you again for your time and consideration.
> >
> > The Authors

---

### Author Response · Authors · 2025-11-19
**General response to reviewers**

We thank all reviewers for their constructive and insightful feedback. We are encouraged that reviewers recognized our work as a **timely and novel** contribution to the emerging MM-DiT architecture (Reviewer kx3n, ruF6, xRya), praised the **principled solution** of replacing implicit point matching with explicit correspondence maps (Reviewer fc7b, kx3n, xRya, FWPs), and acknowledged our **state-of-the-art** performance on DragBench (Reviewer fc7b, kx3n, ruF6, xRya, FWPs).

Our work has benefited substantially from your feedback. Below we summarize the main modifications to the manuscript (highlighted in blue in the revised PDF):
1. **Runtime & Memory Analysis**. Addressing the concerns of Reviewers fc7b, kx3n, ruF6, and FWPs, we provided a detailed breakdown of inference latency and memory usage in Appendix B.4. We introduced a "Normal Generation" internal baseline to rigorously quantify the relative overhead and added an "Optimized" setting to demonstrate that our method achieves interactive speeds competitive with TTO-free baselines.
2. **Clarifications on Contributions & Details**. In response to Reviewers kx3n and ruF6, we refined our contribution statements to align with the method. We also expanded the experimental details to provide a more comprehensive description of settings.
3. **Expanded User Study**. Following Reviewer FWPs' suggestion, we significantly increased the number of expert participants from 20 to 32 (updated in Table 2). This sample size exceeds the protocols of prior baselines (e.g., GoodDrag), ensuring greater statistical stability and further confirming the human preference for our method.

We hope the provided revisions and explanations adequately resolve the issues highlighted in the reviews, and we welcome any additional questions. If you find our response satisfactory, we respectfully invite you to consider raising your scores in recognition of our core contribution: the first training-free, MM-DiT–based attention-control method enabling stable, high-fidelity drag editing without test-time optimization.

---

### Author Response · Authors · 2025-12-01
**Summary of Rebuttal and Discussion**

Dear Area Chair,

We would like to briefly summarize our rebuttal and discussion to assist your final recommendation. All reviewers recognized our work as a **timely and novel** contribution to the emerging MM-DiT architecture (Reviewer kx3n, ruF6, xRya), praised the **principled solution** of replacing implicit point matching with explicit correspondence maps (Reviewer fc7b, kx3n, xRya, FWPs), and acknowledged our **state-of-the-art** performance on DragBench (Reviewer fc7b, kx3n, ruF6, xRya, FWPs).

During the discussion, we extensively updated the manuscript to address all concerns. Notably, Reviewer kx3n explicitly wrote: ***“It helped clarify some concerns. I will maintain my recommendation for acceptance.”***

Here is a summary of how we resolved key issues and significantly elevated the paper's contribution:

1. **Comparative Results Confounded by Backbone Differences (Reviewers fc7b, kx3n, xRya, FWPs)**
   We clarified that the Explicit Correspondence Map is tightly coupled to MM-DiT’s RoPE-based attention and text–vision fusion, and is not a trivial “stronger backbone” change from U-Net. On the *same* MM-DiT backbone, removing our correspondence-driven attention/preservation (“– D Pres., – Attn Refine”) leads to clear drops in identity and drag accuracy (Table 3, Figure 6). In Appendix B.3, porting WTA + Latent Init to FastDrag (U-Net) still yields consistent gains, isolating LazyDrag’s algorithmic benefits.

2. **Runtime & Efficiency (Reviewers fc7b, kx3n, ruF6)**
   Appendix B.4 now reports detailed timing and memory on an H800 GPU for FastDrag, DragText, a Normal MM-DiT generation baseline, and our method. In the optimized setting, LazyDrag achieves **4.31 ± 0.67 s** total latency, comparable to FastDrag (**4.21 ± 0.39 s**) and much faster than DragText (**27.88 ± 9.04 s**) even under a much larger backbone. We also highlight that inversion is performed once and reused across edits, so the per-edit cost remains low in interactive workflows.

3. **Novelty vs. TTO-Free / Motion-Guided Methods (Reviewers ruF6, xRya, FWPs)**
   We clarified that prior methods remain constrained by unstable *implicit* attention-based matching, which forces low-strength inversion or TTO and limits inpainting/text guidance. LazyDrag points out this core problem and replaces this implicit matching with an *explicit* correspondence map at the attention level, enabling stable full-strength inversion without TTO. We distinguish our fully training-free design from InstantDrag/LightningDrag (which require additional training), and show via Table 5 / Figure 11 that latent warping alone (as in FastDrag) is insufficient without our correspondence-driven attention.

4. **Full-Strength Inversion, Metrics, and Contribution Alignment (Reviewers kx3n, ruF6)**
   For Reviewer kx3n, we clarified that full-strength inversion is a *capability*, not a requirement: prior methods avoid it because implicit matching becomes unstable; our explicit correspondence allows full strength when stronger inpainting or text guidance is needed, while still allowing lower strengths for speed. For evaluation (kx3n, ruF6), we justified Semantic Consistency (SC) and Perceptual Quality (PQ) as standard, LLM-based and perceptual metrics widely used in recent editing work, and explained why IF is excluded due to its known unreliability in drag editing. We also refined the contribution statements to align clearly with the two-stage method as requested.

5. **Robustness, Hyperparameters, and WTA Behavior (Reviewers fc7b, xRya)**
   For robustness and activation timestep, we clarified to Reviewers fc7b and xRya that all benchmark results use a single global activation timestep of **40** without per-edit tuning. Under this fixed setting, LazyDrag already outperforms all baselines and remains tolerant to moderately misplaced drag points. We explained the inherent trade-off between precision and naturalness and described how WTA behaves in dense-drag regions, noting the inherent ~16-pixel tolerance due to VAE + patchification and documenting the failure modes at extreme settings.

6. **User Study Scale and Statistical Reliability (Reviewer FWPs)**
   Responding to Reviewer FWPs, we increased the user study from **20 to 32 expert participants**, each evaluating 32 samples. This exceeds comparable drag-editing user studies (e.g., GoodDrag) and strengthens the statistical reliability of the human preference results in favor of LazyDrag.

We hope this concise mapping from concerns to concrete changes and analyses will be helpful in your decision.

Best regards,
The Authors

---

### Meta-Review · Area_Chair_GJnU · 2026-01-07

**Summary:**

The reviewers agree the paper is well written, conceptually clear, and delivers strong empirical improvements on DragBench and user studies. However, they note that (i) efficiency claims lack quantitative runtime/memory evidence, (ii) some gains may stem from using a stronger MM-DiT backbone rather than the editing strategy alone, (iii) novelty is somewhat incremental relative to prior drag-editing pipelines, and (iv) generalization and evaluation breadth are limited. Despite these concerns, the method is viewed as a solid and meaningful advance with clear practical value.

**Reviewer Concerns:**

As far as I can tell, the reviewers’ comments have been adequately addressed. The authors also raised concerns regarding the reliability of Reviewer xRya’s comments. Based on my own assessment, I believe these concerns are valid. Accordingly, I have discounted the comments from this reviewer in my evaluation.

**Reviewer Scores:**

Reviewer fc7b: 6
Reviewer kx3n: 6
Reviewer ruF6: 6
Reviewer xRya: 4
Reviewer FWPs: 6

I expect reviewers fc7b, kx3n, ruF6, FWPs will maintain their original scores. Reviewer xRya will likely increase the score to 5 or 6.

---

### Decision · Program_Chairs · 2026-01-26

Accept (Poster)